# PRISM: Privacy-preserving Improved Stochastic Masking for Federated Generative Models

## Abstract

While training generative models in distributed settings has recently become increasingly important, prior efforts often suffer from compromised performance, increased communication costs, and privacy issue. To tackle these challenges, we propose PRISM: a new federated framework tailored for generative models that emphasizes not only strong and stable performance but also resource efficiency and privacy preservation. The key of our method is to search for an optimal stochastic binary mask for a random network rather than updating the model weights; *i.e.*, identifying a "strong lottery ticket": a sparse subnetwork with competitive generative performance. By communicating the binary mask in a stochastic manner, PRISM minimizes communication overhead while guaranteeing differential-privacy (DP). Unlike traditional GAN-based frameworks, PRISM employs the maximum mean discrepancy (MMD) loss, ensuring stable and strong generative capability, even in data-heterogeneous scenarios. Combined with our weight initialization strategy, PRISM also yields an exceptionally lightweight final model with no extra pruning or quantization, ideal for environments such as edge devices. We also provide a hybrid aggregation strategy, PRISM-$\alpha$, which can trade off generative performance against communication cost. Experimental results on MNIST, CelebA, and CIFAR10 demonstrate that PRISM outperforms the previous methods in both IID and non-IID cases, all while preserving privacy at the lowest communication cost. To our knowledge, we are the first to successfully generate images in CelebA and CIFAR10 with distributed and privacy-considered settings. Our code is available at PRISM.

## 1 Introduction

Recent generative models have demonstrated remarkable advancements in image quality and have been widely extended to various domains, including image-to-image translation Isola et al. (2017); Saharia et al. (2022), image editing Abdal et al. (2019); Tov et al. (2021), text-to-image generation Rombach et al. (2022); Ramesh et al. (2022), and video generation Skorokhodov et al. (2022). To reach a certain level of image quality of generative models, a significant volume of training data is generally required. However, in many applications, data samples are often distributed across different client devices and should be kept private (*e.g.*, personal data at smartphones), posing challenges for centralized training. Federated learning (FL) McMahan et al. (2017) is a promising paradigm tailored to this setup, enabling clients to collaboratively train a global model based on repeated local updates and server-side model aggregation, without sharing each client's local dataset to the third party. One of the major bottlenecks in FL is the significant communication cost for exchanging the model between the server and the clients. Moreover, it is shown that each client's local dataset can be inferred from its local model transmitted to the server, resulting in privacy issues Zhu et al. (2019).

A few recent works have specifically focused on training generative models over distributed clients Hardy et al. (2019); Rasouli et al. (2020); Li et al. (2022); Zhang et al. (2021); Amalan et al. (2022). These methods are generally built upon generative adversarial networks (GANs) Goodfellow et al. (2020), which have shown impressive results in the field of image generation. Based on GANs, prior works such as DP-FedAvgGAN Augenstein et al. (2019), GS-WGAN Chen et al. (2020), and Private-FLGAN Xin et al. (2020) apply differential privacy (DP) Dwork et al. (2006); Mironov (2017) to mitigate the potential privacy risk in FL setups. However, existing works still face several challenges: 1) Due to the training instability of GANs Farnia & Ozdaglar (2020a;b);

Wang et al. (2022), previous approaches underperform, especially in non-IID (independent, identically distributed) data distribution scenarios with strong data heterogeneity across FL clients. 2) Performance evaluations are limited to relatively simple datasets such as MNIST, Fashion MNIST, and EMNIST. 3) They suffer from a significant communication overhead during model exchanges between the server and clients.

To overcome the above challenges, we propose PRivacy-preserving Improved Stochastic-Masking for generative models (PRISM), a new strategy for training generative models in FL settings. At the heart of PRISM lies the strong lottery ticket (SLT) Frankle & Carbin (2018), a subnetwork in a randomly initialized network that can achieve strong performance. Our approach seamlessly integrates the Edge-Popup (EP) method Ramanujan et al. (2020); Yeo et al. (2023), a pioneering algorithm crafted to discern a supermask in densely interconnected networks. By harnessing the SLT in the FL paradigm, our focus shifts to finding an optimal global binary mask, while keeping initialized weights static. This enables each client to directly transfer the binary mask with the server instead of the full model, significantly reducing the overload in each communication round. In addition, due to the stochastic binary masks that are randomly sampled from the Bernoulli distribution, PRISM also provides DP-guarantee. We also opt MMD loss during the client-side local update process to stably find the SLT for generative models. By doing so, PRISM exhibits consistent and robust performance even in non-IID FL settings, unlike traditional GAN-based approaches. When training is finished, PRISM produces a lightweight final model as each weight is already quantized thanks to our initialization strategy, providing significant advantages for resource-constrained edge devices. Finally, we propose a variation of our method, termed PRISM-$\alpha$, that can strategically control the amount of binary mask communication and score communication. By letting $\alpha$ be the portion of the score communication layer, PRISM-$\alpha$ explores the trade-off between communication cost and image quality depending on the application and resource constraint.

We provide qualitative and qualitative comparisons using MNIST, CelebA, and CIFAR10 datasets and confirm that PRISM and PRISM-$\alpha$ present remarkable performance improvements against the traditional GAN-based approaches with less communication costs, in both IID and non-IID data distribution settings. PRISM expands the applicability of privacy-aware federated generative models to CelebA and CIFAR10 datasets, producing clean images on these datasets for the first time. We also demonstrate that PRISM enables edge devices to save resources not only during training but also during the inference stage.

Overall, our main contributions can be summarized as follows:

- We propose PRISM, a SLT-based FL framework that leverages stochastic binary masks to facilitate communication-efficient FL for training generative models while providing DP-guarantee. In conjunction with the weight initialization strategy, a lightweight final model is produced when training is finished.

- We make a breakthrough against instability and performance deficiencies by introducing the MMD loss to federated generative models. Our objective function ensures consistent convergence and enhanced performance in both IID and non-IID FL scenarios.

- We present a hybrid score and binary mask aggregation strategy termed PRISM-$\alpha$, to control the trade-off between communication and model capability. PRISM-$\alpha$ is able to further improve the performance while incurring a small amount of additional communication cost.

To the best of our knowledge, this is the first work to address the challenges on communication efficiency, privacy, and performance instability altogether for federated generative models. We provide new directions to this area based on several unique characteristics, including SLT with stochastic binary mask, MMD loss, and hybrid score/mask communications.

## 2    RELATED WORK

**Federated learning.** FL has recently achieved a significant success in training a global model in a distributed setup, eliminating the necessity of sharing individual client's local datasets with either the server or other clients. Research has been conducted for various aspects in FL such as data heterogeneity Zhao et al. (2018); Li et al. (2021b), communication efficiency Isik et al. (2022); Li et al. (2021a); Mitchell et al. (2022); Basat et al. (2022), privacy Wei et al. (2020), with most of them focusing on the image classification task. Related to our approach, Isik et al. (2022); Li et al. (2021a) adopted binary mask communication to reduce the communication cost in FL. Li et al. (2021a)

introduces binary mask communication, focusing communication efficiency and personalization in decentralized environments. Isik et al. (2022) utilizes stochastic masks to minimize uplink overhead and proposes a bayesian aggregation method to robustly manage scenarios with partial client participation. In connection with these methods, PRISM also employs binary masks for lightweight communication between the client and the server. To distinguish our work from previous works, we would like to clarify our specific objectives. Tailored to FL for generative models, we introduce a stable MMD loss Gretton et al. (2006; 2012) and a hybrid method that strikes an efficient balance between performance and cost. Additionally, we conduct extensive experiments to validate these contributions in the realms of image generation and communication cost.

**FL for generative models.** Several recent works have aimed to incorporate generative models into distributed settings Hardy et al. (2019); Amalan et al. (2022); Li et al. (2022); Zhang et al. (2021); Rasouli et al. (2020); Augenstein et al. (2019); Chen et al. (2020); Xin et al. (2020). MD-GAN Hardy et al. (2019) was the first attempt to apply generative models in the FL framework using GANs Goodfellow et al. (2020), which have been extensively studied in image generation tasks. In MD-GAN, each client holds a discriminator, and the server aggregates the discriminator feedback received from each client to train the global generator. To prevent overfitting of local discriminators, clients exchange discriminators, incurring additional communication costs. Multi-FLGAN Amalan et al. (2022) proposed all vs all game approach by employing multiple generators and multiple discriminators and then selecting the most powerful network, to enhance the model performance. IFL-GAN Li et al. (2022) improves both performance and stability by weighting each client's feedback based on the MMD between the images generated by the global model and the local generator. This approach maintains a balance between the generator and the discriminator, which leads to Nash Equilibrium. Other works such as Zhang et al. (2021); Rasouli et al. (2020) have also explored the utilization of GANs in FL. However, these works do not consider the challenge of privacy preservation in the context of FL, and suffer from resource issues during training and inference.

Only a few prior works have focused on the privacy issue in federated generative models Augenstein et al. (2019); Chen et al. (2020). DP-FedAvgGAN Augenstein et al. (2019) introduces a framework that combines federated generative models and differential privacy (DP) Dwork et al. (2006); Mironov (2017) to ensure privacy preservation. GS-WGAN Chen et al. (2020) adopts Wasserstein GAN Gulrajani et al. (2017) to bypass the cumbersome challenge of searching for an appropriate DP-value, leveraging the Lipshitz property. While these approaches have successfully integrated FL and generative models, they inherit drawbacks such as notorious instability of GANs Farnia & Ozdaglar (2020a;b); Wang et al. (2022) and face issues with data heterogeneity. Moreover, all existing approaches suffer from significant communication cost during training and storage cost during inference. Our proposed PRISM addresses all the above issues of prior federated generative model works, with several unique characteristics including SLT Frankle & Carbin (2018) with stochastic binary mask, MMD loss Gretton et al. (2006; 2012), and hybrid score/mask communications.

## 3 BACKGROUND

### 3.1 STRONG LOTTERY TICKETS

Strong Lottery Ticket (SLT) Frankle & Carbin (2018); Malach et al. (2020); Orseau et al. (2020) is a hypothesis related to pruning of dense neural networks, claiming that a randomly initialized network already contains a sparse subnetwork that achieves a superior performance. To this end, Edge-popup (EP) algorithm Ramanujan et al. (2020) was proposed to discover supermask within the dense network. The EP algorithm introduces a scoring mechanism to select potentially important weights among the widespread initialized weight values. The operation of EP algorithm unfolds as follows: Given a randomly initialized dense network $W_{init}$, a learnable score $s$ is trained while keeping the weight values in frozen. These scores are designed to encapsulate the importance of each weight for the objective function. As the scores get iteratively updated, the EP algorithm progressively shrinks the model by applying binary masks to weights with higher scores, indicating their potentials to be included in the winning lottery ticket. The obtained SLT can be expressed as $W = W_{init} \odot M$, where $M$ is the obtained binary mask and $\odot$ denotes element-wise multiplication. In Yeo et al. (2023), the existence of winning tickets in generative models is shown based on the MMD loss.

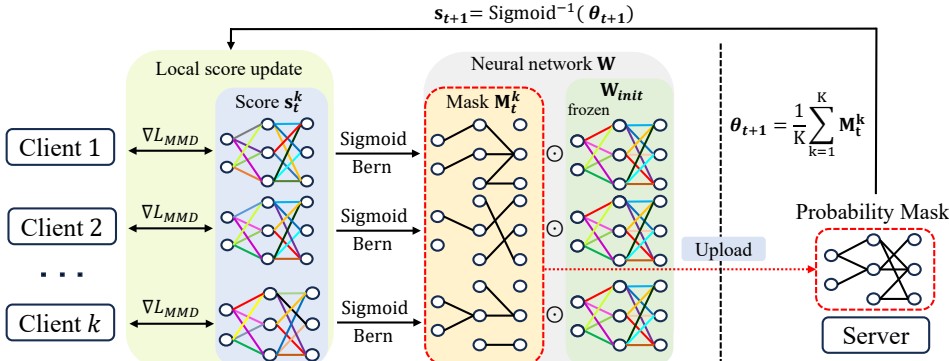

Figure 1: **Overview of PRISM.** PRISM finds the supermask for generative models in a FL scenario. At every round $t$, each client $k$ updates a local score $s_t^k$ via MMD loss and generates the binary mask $M_t^k$, which is sent to the server. The server aggregates the masks and use $\theta_{t+1}$ to estimate the global probability, which is converted to a score $s_{t+1}$ and broadcasted to the clients for the next round.

## 4 METHOD

We consider a FL setup with $K$ clients, where each client $k$ has its own local dataset $\mathcal{D}^k$. Starting from a randomly initialized model $W_{init}$, the clients aim to collaboratively obtain a global generative model $W^*$ that well-reflects all data samples in the system, i.e., in $\cup_{k=1}^{K} \mathcal{D}^k$.

**Overview of approach.** We propose PRISM, a new FL framework tailored to generative models that can handle communication, model performance, and privacy issues of prior works. Inspired by SLT hypothesis, PRISM aims to find a good subnetwork from the initialized generative model $W_{init}$, and adopts the final SLT as the global model. To this end, we shift our focus to finding an optimal binary mask $M^*$ that has either 1 or 0 in its element, and construct the final global model as $W^* = W_{init} \odot M^*$. Figure 1 shows the overview of our PRISM. At a high-level, each client $k$ generates a binary mask $M_t^k$ based on its local dataset at every communication round $t$, which is aggregated at the server. After repeating the process for multiple rounds $t = 1, 2, \ldots, T$, PRISM produces the final supermask $M^* = M_T$. In the following, we describe the detailed training procedure of PRISM along with its advantages.

### 4.1 PRISM : PRIVACY-PRESERVING IMPROVED STOCHASTIC MASKING

**Local score updates with MMD loss.** Before training starts, the server randomly initializes the model $W_{init}$ and broadcasts it to all clients, which is frozen throughout the training process. In the beginning of each round $t$, all clients download the score vector $s_t$ from the server, which represents the importance of each parameter in $W_{init}$. Intuitively, if the score value of a specific parameter is high, the corresponding weight is more likely be included in the final SLT. PRISM lets each client $k$ to update the score vector $s_t$ based on its local dataset to obtain $s_t^k$, which will be used to generate the local mask. In this local score update procedure, we leverage maximum mean discrepancy (MMD) loss Gretton et al. (2006; 2012), which provides stable convergence for training generative models. As in Santos et al. (2019); Ramanujan et al. (2020), we take VGGNet pretrained on ImageNet as a powerful characteristic kernel. Specifically, given the local dataset $\mathcal{D}^k = \{x_i^k\}_{i=1}^{N}$ of client $k$ and the fake image set $\mathcal{D}_{fake}^k = \{y_i^k\}_{i=1}^{M}$ produced by its own generator, the local objective function at each client $k$ is written as follows:

$$\mathcal{L}_{MMD}^k = \left\| \mathbb{E}_{x \sim \mathcal{D}^k}[\psi(x)] - \mathbb{E}_{y \sim \mathcal{D}_{fake}^k}[\psi(y)] \right\|^2 + \left\| \mathrm{Cov}(\psi(\mathcal{D}^k)) - \mathrm{Cov}(\psi(\mathcal{D}_{fake}^k)) \right\|^2, \quad (1)$$

where $\psi(\cdot)$ is a function that maps each sample to the VGG embedding space. Specifically, each client aims to match the mean and covariance between real and fake samples after mapping them to the VGG embedding space using kernel $\psi(\cdot)$. Based on Eq. 1, each client locally updates the scores to minimize the MMD loss according to $s_t^k = s_t - \eta \nabla \mathcal{L}_{MMD}^k$. Here, we note that the VGG network is utilized only for computing the MMD loss, and is discarded when training is finished.

We would like to highlight that taking advantage of MMD loss guarantees stable training for finding the SLT of the generative model. This enables PRISM to handle the issue of prior federated generative model approaches mainly adopting GANs that suffer from unstable training and limited performance in data heterogeneous FL settings.

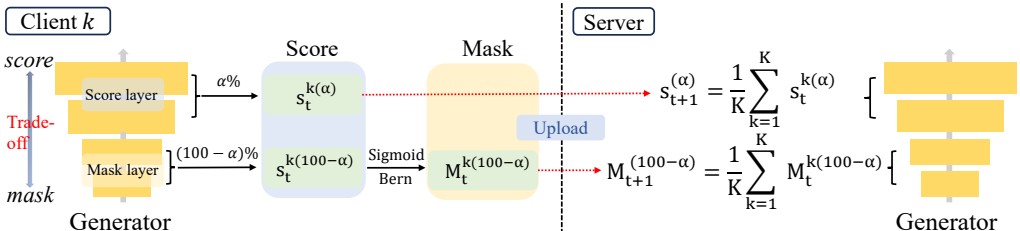

Figure 2: **Overview of PRISM-$\alpha$.** With a controllable parameter $\alpha$, *score layers* are selected from $\alpha\%$ of layers and *mask layers* are chosen from the remaining layers. At communication round $t$, each client $k$ uploads the learned scores $s_t^{k(\alpha)}$ from the *score layers* and binary masks $M_t^{k(1-\alpha)}$ from the *mask layers*. Then, the server aggregates the received scores and binary masks, respectively.

**Binary mask generation and aggregation.** After the local score update process, each client $k$ maps the score $s_t^k$ to a probability value $\theta_t^k \in [0, 1]$ as $\theta_t^k = Sigmoid(s_t^k)$, where $Sigmoid(\cdot)$ is the sigmoid function. The obtained $\theta_t^k$ is then used as the parameter of the Bernoulli distribution to generate the stochastic binary mask $M_t^k$, according to $M_t^k \sim Bern(\theta_t^k)$. Each client $k$ uploads only this binary mask $M_t^k$ to the server, significantly reducing the communication overhead. The stochasticity of the mask also enables PRISM to preserve privacy. At the server-side, the received masks are aggregated to estimate the global Bernoulli parameter as $\theta_{t+1} = \frac{1}{K} \sum_{k=1}^{K} M_t^k$, which can be interpreted as the probabilistic score that reflects the importance of the overall clients' weights. $\theta_{t+1}$ is then converted to the score through the inverse of the sigmoid function according to $s_{t+1} = Sigmoid^{-1}(\theta_{t+1})$, which is broadcasted to the clients in the beginning of next round.

**Inference-time storage efficiency.** When training is finished after $T$ rounds of FL, $\theta_T$ is obtained at the server. The supermask is then generated following $M^* \sim Bern(\theta_T)$, which is used to obtain the final global model as $W^* = W_{init} \odot M^*$. This final model $W^*$ can be stored efficiently even in resource-constrained edge devices, due to the model initialization strategy. When initializing $W_{init}$ in PRISM, we employ the standard deviation of Kaiming Normal distribution He et al. (2015), which means that the weight value in layer $l$ is sampled from $\{-\sqrt{2/n_{l-1}}, \sqrt{2/n_{l-1}}\}$. Hence, by storing the scaling factor $\sqrt{2/n_{l-1}}$, each parameter in the initial model $W_{init}$ is already quantized to a 1-bit value. This makes the final model exceptionally lightweight without extra pruning or quantization, which will be also confirmed via comparison in Section 5.4.

### 4.2   HYBRID SCORE AND MASK AGGREGATION

PRISM in Section 4.1 achieves minimum communication cost by letting clients transmit only the binary mask during FL. In this subsection, we additionally introduce PRISM-$\alpha$, which can flexibly control the trade-off between communication overhead and performance depending on the application and resource constraint. Figure 2 illustrates the idea of PRISM-$\alpha$. Instead of sending the full binary mask to the server, every client $k$ sends the deterministic score for the fixed $\alpha\%$ of layers (i.e., score layers) and the binary mask for the remaining $(100 - \alpha)\%$ of layers (i.e., mask layers). This enables the server to more accurately obtain the global score, as the score layers do not have any stochasticity. This further leads to improved model performance, which is an advantage of PRISM-$\alpha$ that can be achieved via additional communication cost for sending the non-binary values. At the server side, the score layers are aggregated according to $s_{t+1}^{(\alpha)} = \frac{1}{K} \sum_{k=1}^{K} s_t^{k(\alpha)}$, while the mask layers are aggregated as $\theta_{t+1}^{(100-\alpha)} = \frac{1}{K} \sum_{k=1}^{K} M_t^{k(100-\alpha)}$. The obtained score is directly broadcasted to the clients, while the aggregated mask is converted to the score as in PRISM before broadcasting. Note that as mask layers consistently transmits stochastic outcomes, PRISM-$\alpha$ still protects privacy. Even in cases where the server can infer local data from the score layers, this can be easily handled by applying secure aggregation Bonawitz et al. (2016) to the score layers.

### 4.3   PRIVACY PRESERVATION

Differential privacy (DP) Dwork et al. (2006), Rényi-Differential privacy (RDP) Mironov (2017) are commonly adopted metrics in FL society to prevent data being inferred from the model. As discussed in Section **??**, random sampling from distribution family provides privacy amplification. PRISM also enjoys the binary symmetric RDP thanks to the stochasticity of Bernoulli sampling, directly following the derivations of Isik et al. (2022); Imola & Chaudhuri (2021).

Table 1: **Quantitative comparison in IID scenario.** We compare the FID, P&R, D&C, communication cost. Communication cost is reported by capturing the number of bytes exchanged from the client to the server. We set $\alpha = 70$ for PRISM-$\alpha$ while $\alpha = 100$ for PRISM-$\alpha$ (full score). Since MD-GAN and PRISM-$\alpha$ (full score) do not provide privacy protection, they are considered as upper bounds. Although GS-WGAN can achieve highest precision and density in CIFAR10, it fails to produce reasonable images as shown in Figure 3.

| Method (comm.cost) | Privacy | Metric | MNIST | CelebA | CIFAR10 |
|---|---|---|---|---|---|
| MD-GAN (14MB) | ✗ | **FID** ↓ | 5.5364 | 6.74 | 28.5932 |
| | | **P&R** ↑ | 0.8081 / 0.7414 | 0.796 / 0.6243 | 0.7949 / 0.5613 |
| | | **D&C** ↑ | 0.7525 / 0.7955 | 1.05 / 0.8919 | 1.071 / 0.5512 |
| PRISM-100 (full score) (24MB) | ✗ | **FID** ↓ | 3.6597 | 11.0592 | 36.9452 |
| | | **P&R** ↑ | 0.8001 / 0.823 | 0.7824 / 0.4229 | 0.6602 / 0.3706 |
| | | **D&C** ↑ | 0.7312 / 0.8382 | 1.0275 / 0.8221 | 0.5751 / 0.4557 |
| GS-WGAN (15MB) | ✓ | **FID** ↓ | 68.53 | 203.6972 | 193.5444 |
| | | **P&R** ↑ | 0.0975 / 0.0794 | 0.1675 / 0 | **0.9961** / 0 |
| | | **D&C** ↑ | 0.0257 / 0.0367 | 0.0357 / 0.002 | **1.4743** / 0.0541 |
| DP-FedAvgGAN (14MB) | ✓ | **FID** ↓ | 87.9032 | 206.0260 | 195.9148 |
| | | **P&R** ↑ | 0.2237 / 0.0688 | 0.0518 / 0.0585 | 0.6671 / 0.0039 |
| | | **D&C** ↑ | 0.0549 / 0.0225 | 0.0111 / 0.0017 | 0.2302 / 0.0302 |
| PRISM (5.75MB) | ✓ | **FID** ↓ | 8.778 | 18.1198 | 62.373 |
| | | **P&R** ↑ | 0.682 / 0.7255 | 0.772 / 0.2834 | 0.7439 / 0.0774 |
| | | **D&C** ↑ | 0.4584 / 0.6063 | 0.9281 / 0.6995 | 0.8234 / 0.3159 |
| PRISM-70 (9.6MB) | ✓ | **FID** ↓ | **4.6893** | **13.1921** | **50.682** |
| | | **P&R** ↑ | **0.8001 / 0.823** | **0.7858 / 0.3926** | 0.685 / **0.256** |
| | | **D&C** ↑ | **0.731 / 0.838** | **0.978 / 0.779** | 0.6066 / **0.3796** |

## 5 EXPERIMENTS

In this section, we validate the effectiveness of PRISM on MNIST, CelebA, and CIFAR10 datasets. The training set of each dataset is distributed across 10 clients following either IID or non-IID data distributions, where the details are described in each subsection. We set $\alpha = 70$ for our PRISM-$\alpha$ regardless of datasets. The effect of varying $\alpha$ on PRISM-$\alpha$ is also studied in Section 5.3.

**Baselines.** We compare our method with three previous approaches on federated generative models: DP-FedAvgGAN Augenstein et al. (2019) and GS-WGAN Chen et al. (2020) that preserve privacy, and MD-GAN Hardy et al. (2019) which does not. As MD-GAN does not consider privacy, we employ it as an upper bound for performance comparison.

**Performance metrics.** We evaluate the generative performance of each scheme using the commonly adopted metrics, including Fréchet Inception Distance (FID) Heusel et al. (2017), Precision & Recall Kynkäänniemi et al. (2019), Density & Coverage Naeem et al. (2020). We further demonstrate the efficiency of PRISM by comparing the required communication cost (MB) at each FL round. Finally, we compare the storage size (MB) of different schemes during the inference stage.

### 5.1 IID CASE

In this subsection, we consider an IID scenario where the training set of each dataset is distributed to clients uniformly at random. We first compare various evaluation metrics of different schemes (Table 1). It can be seen that PRISM outperforms traditional GAN-based models while requiring significantly lower communication cost. In addition, our proposed hybrid strategy PRISM-$\alpha$ exhibits notable performance enhancements by trading off performance against communication cost. In CIFAR10, while GS-WGAN achieves highest performance in specific metrics (precision and density), it fails to generate reasonable images as shown in Figure 3, which depicts qualitative results of generated images. From Figure 3, we observe that existing methods that preserve privacy tend to generate distorted images in both CelebA and CIFAR10, while our method produces high quality results. Although the generated images are not perfect for CIFAR10, our approach still produces better quality of images compared with the privacy-preserving baselines. In Figure 4(a), we also report the relationship between FID, number of parameters in the generator, and communication cost of each scheme. Here, for a fair comparison with the baselines, we reduce the number of generator parameters from 6.3M to 3.5M, and denote by PRISM (small). It can be seen that PRISM consistently

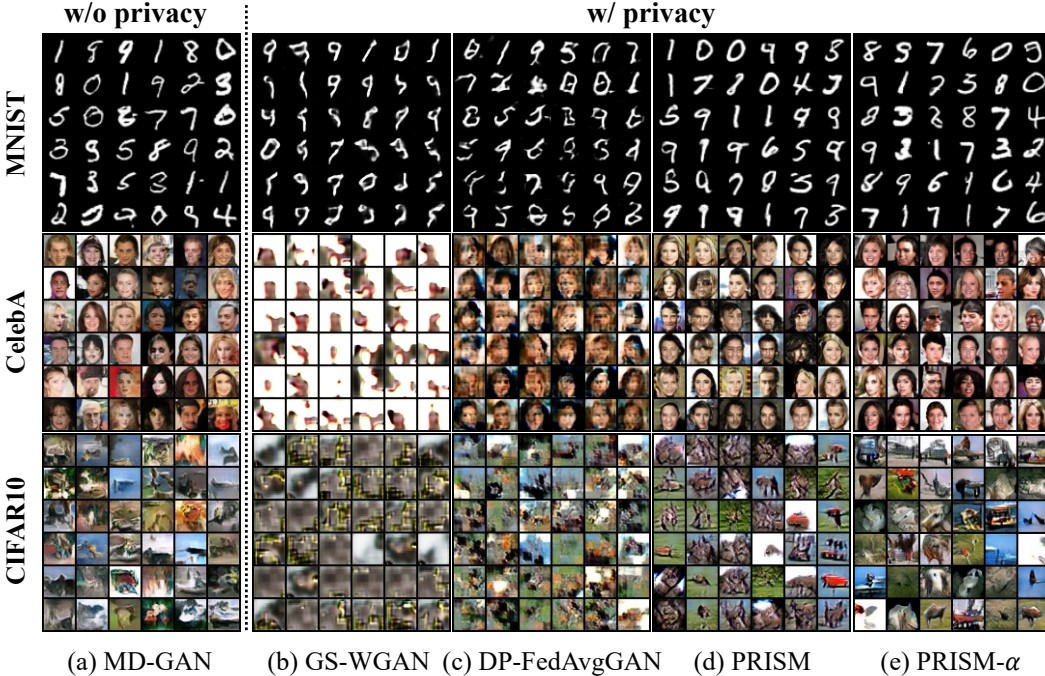

Figure 3: **Qualitative results in IID scenario.** We compare the generated images from the models in Table 1 on MNIST, CelebA, and CIFAR10. Here, $\alpha = 70$ for PRISM-$\alpha$.

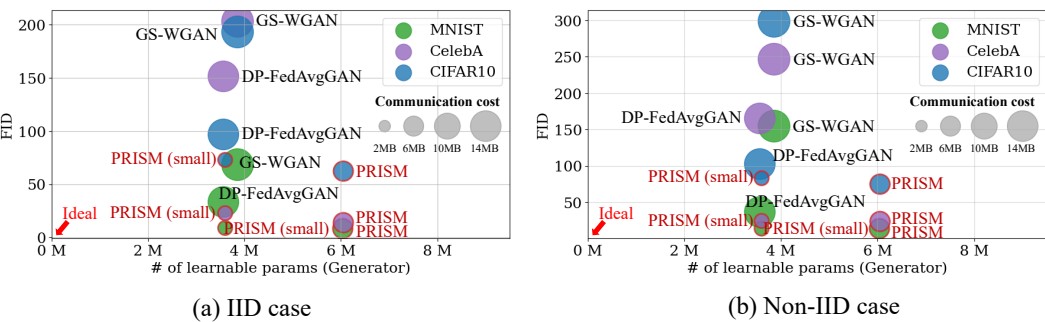

(a) IID case             (b) Non-IID case

Figure 4: **The performance of baselines and our PRISM.** X-axis represents the number of learnable parameters of generator, while Y-axis represents FID. The diameter of each circle denotes the required communication cost at every round. The ideal case is the bottom-left corner with a small diameter. For a fair comparison and to validate the training stability of PRISM, we reduce the capacity of PRISM by half (to match the capacity of other models) and denote by PRISM (small).

demonstrates stable performance while offering more lightweight communication cost, confirming its significant advantage.

## 5.2 NON-IID CASE

In this subsection, we investigate another practical yet challenging scenario, the non-IID case, where the local data distributions across clients are different. Since the clients have heterogeneous data distributions in this setup, achieving sufficient performance becomes more challenging. We sort MNIST and CIFAR10 datasets by class labels and divide into 40 partitions. Then, we randomly assign four partitions to each client. As CelebA has multiple attributes in a single image, we split the overall dataset into two sets that contain opposite attributes (male and female) and distribute each to 5 clients, to model the non-IID scenario. Table 2 shows the quantitative comparison of baselines and our methods. PRISM and PRISM-$\alpha$ exhibit robust performance in non-IID case, even better than MD-GAN (no privacy) on MNIST. Figure 5 shows that despite data heterogeneous nature, our methods successfully generate high quality images, while traditional methods exhibit subpar quality on MNIST and CelebA, with GS-WGAN even failing on MNIST. In Figure 4(b), we visualize the FID, number of parameters in the generator, and communication cost on non-IID scenario. It can

Table 2: **Quantitative comparison in non-IID scenario.** We compare the FID, P&R, D&C, and communication cost. We report the number of bytes exchanged from the client to the server as communication cost. As in Table 1, we set $\alpha = 70$ for PRISM-$\alpha$, while $\alpha = 100$ for PRISM-$\alpha$ (full score). MD-GAN and PRISM-$\alpha$ (full score) do not preserve privacy.

| Method (comm.cost) | privacy | Metric | MNIST | CelebA | CIFAR10 |
|---|---|---|---|---|---|
| MD-GAN (14MB) | ✗ | **FID** ↓ | 40.4037 | 18.5903 | 51.2847 |
| | | **P&R** ↑ | 0.3297 / 0.491 | 0.715 / 0.4673 | 0.827 / 0.1968 |
| | | **D&C** ↑ | 0.1145 / 0.1434 | 0.7363 / 0.6793 | 1.2201 / 0.3829 |
| PRISM-100 (full score) (24MB) | ✗ | **FID** ↓ | 7.7762 | 14.1072 | 52.8479 |
| | | **P&R** ↑ | 0.7616 / 0.7387 | 0.782 / 0.35 | 0.6585 / 0.2749 |
| | | **D&C** ↑ | 0.5895 / 1.0453 | 0.7217 / 0.2834 | 0.5506 / 0.3464 |
| GS-WGAN (15MB) | ✓ | **FID** ↓ | 155.2829 | 246.682 | 299.27 |
| | | **P&R** ↑ | 0.0216 / 0.0011 | 0.2318 / 0.0 | 0.7097 / 0 |
| | | **D&C** ↑ | 0.0044 / 0.0009 | 0.0566 / 0.0015 | 0.6558 / 0.0504 |
| DP-FedAvgGAN (14MB) | ✓ | **FID** ↓ | 131.6076 | 216.2886 | 199.9460 |
| | | **P&R** ↑ | 0.3086 / 0.0608 | 0.0639 / 0.0 | 0.4989 / 0.0007 |
| | | **D&C** ↑ | 0.0674 / 0.0113 | 0.0140 / 0.0015 | 0.2589 / 0.0333 |
| PRISM (5.75MB) | ✓ | **FID** ↓ | 14.4835 | 23.7323 | 75.354 |
| | | **P&R** ↑ | 0.6904 / 0.5599 | 0.7758 / 0.1883 | **0.6805** / 0.086 |
| | | **D&C** ↑ | 0.4627 / 0.4908 | 1.0209 / 0.6082 | **0.5599** / 0.2468 |
| PRISM-70 (9.6MB) | ✓ | **FID** ↓ | **8.1081** | **20.2256** | **54.6813** |
| | | **P&R** ↑ | **0.7378 / 0.7749** | **0.7899 / 0.2391** | 0.6204 / **0.2469** |
| | | **D&C** ↑ | **0.5517 / 0.6505** | **1.0493 / 0.6618** | 0.5006 / **0.3348** |

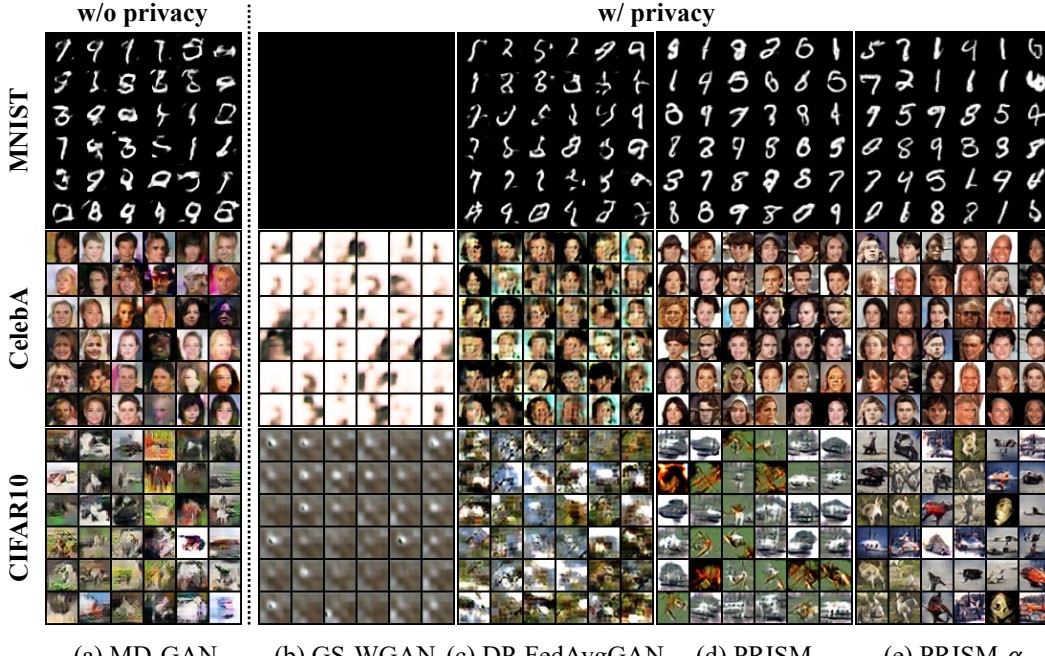

Figure 5: **Qualitative results in non-IID scenario.** We compare the generated images from the models in Table 1 on MNIST, CelebA, and CIFAR10. Here, $\alpha = 70$ for PRISM-$\alpha$. Each client is assigned with four classes in MNIST and CIFAR10, and male or female in CelebA.

be seen that our method achieves similar performance even when the size of the model is reduced, *i.e.,* PRISM (small), demonstrating the stability of PRISM in data-heterogeneous environments. The overall results confirm the advantage of our method compared with existing baselines.

## 5.3 ANALYSIS OF HYBRID AGGREGATION

In this subsection, we further analyze PRISM-$\alpha$, which utilizes both binary mask and score communications. To explore the trade-off between communication cost and generative capability, we consider two strategies: the *backward path* and *forward path*. As illustrated in Figure 2, our default

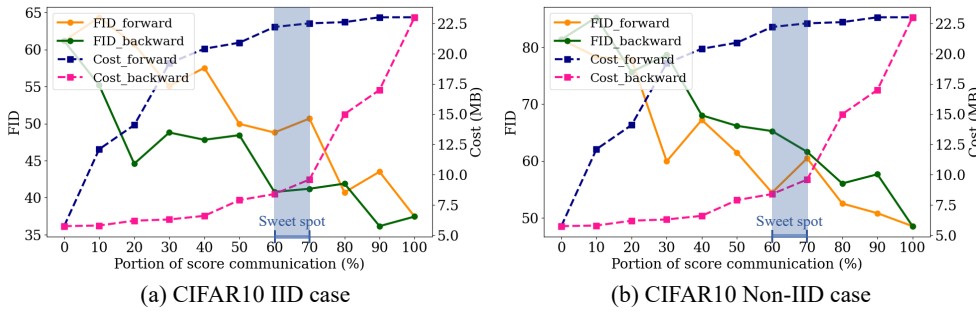

(a) CIFAR10 IID case       (b) CIFAR10 Non-IID case

Figure 6: **Analysis of PRISM-$\alpha$**. The *backward path* selects $\alpha\%$ of *score layers* from deeper layers, while the *forward path* chooses from the opposite layers. Solid-line demonstrates FID following each direction while dash-line shows communication cost (MB) of each path.

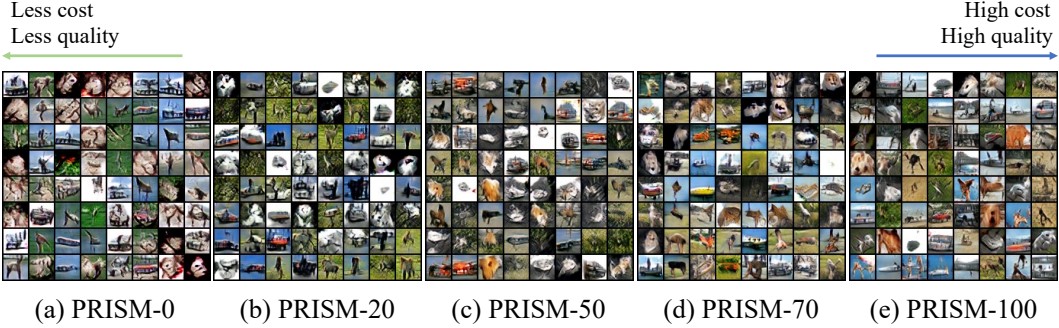

(a) PRISM-0      (b) PRISM-20      (c) PRISM-50      (d) PRISM-70      (e) PRISM-100

Figure 7: **The effect of adjusting $\alpha$.** PRISM-$\alpha$ provides trade-off between communication cost and generative performance. Note that PRISM-0 is identical to PRISM.

configuration, the *backward path* progressively increases the number of *score layers* from deeper layers to earlier layers. Conversely, in the *forward path*, we select *score layers* from earlier layers to deeper layers. Figure 6 visually demonstrates the trade-off between communication cost and FID of both strategies. The FID gradually improves as we increase $\alpha$ values in both cases. Note that the additional communication cost of the *backward path* tends to increase more smoothly. Figure 7 provides comprehensive comparison across a wide range of $\alpha$ values, showing that PRISM-$\alpha$ consistently produces high quality images.

## 5.4 RESOURCE EFFICIENCY IN INFERENCE TIME

Once PRISM identifies the SLT, each client needs to save the final model $W^* = W_{init} \odot M^*$ for inference. As discussed in Section 4.1, one advantage of PRISM is the extremely lightweight final model. This is attributed to the uniform binarization of the weights $W_{init}$ with signed constants, allowing for more efficient storage of each initialized weight through the utilization of ternary quantization Zhu et al. (2016). Table 3 reports the final model size of baselines and our method. While baselines save the full weights, PRISM only stores the pruned and 1-bit quantized values. Note that in addition to our method, applying various lossless compression techniques (*e.g.,* arithmetic coding Rissanen & Langdon (1979)) further reduce the required resources of PRISM.

Table 3: **Final model size.** The storage of PRISM includes the sparse and quantized network and binary mask, while baselines save the full generator.

| Method | Storage |
|---|---|
| MD-GAN | 14 MB |
| GS-WGAN | 15 MB |
| DP-FedAvgGAN | 14 MB |
| PRISM | 7.258MB |

## 6 CONCLUSION

Although image generation task has recently emerged as one of the most promising areas in deep learning, combining federated learning with generative models has not been extensively explored. In this paper, we propose an efficient and stable federated generative framework PRISM that leverages stochastic binary mask and MMD loss. We also introduce a hybrid mask/score aggregation method, PRISM-$\alpha$, which provides flexible and controllable trade-off between the performance and the efficiency. To the best of our knowledge, PRISM is the first framework that consistently generates images with dramatically reduced communication in FL, particularly for CelebA and CIFAR10.

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

# A    PRIVACY

Diffenrential privacy (DP) and Rényi Differential Privacy (RDP) Dwork et al. (2006); Mironov (2017) are the most popular definitions to analysis the privacy in FL environments. These help mitigate privacy concerns by limiting the contribution of individual data points. $(\epsilon, \delta)$-DP and $(\alpha, \epsilon)$-RDP basically calculates the distance of outcome for the algorithm of adjacent datasets.

**Definition 1 (($\epsilon, \delta$)-Differential Privacy)** *A randomized mechanism* $\mathcal{M}$ : $\mathcal{X}$ $\rightarrow$ $\mathcal{R}$ *is* $(\epsilon, \delta)$-*differential privacy, if for any two adjacent datasets* $\mathcal{D}$, $\mathcal{D}'$ *and for any measurable sets* $\mathcal{S}$:

$$\boldsymbol{Pr}[\mathcal{M}(D) \in \mathcal{S}] \le e^{\epsilon} \boldsymbol{Pr}[\mathcal{M}(D') \in \mathcal{S}] + \delta \tag{2}$$

**Definition 2 (($\alpha, \epsilon$) Rényi Differential Privacy)** *For two probability distributions P and Q, the Rényi divergence of order* $\alpha > 1$ *defined as follows:*

$$R_{\alpha}(P||Q) \triangleq \frac{1}{\alpha - 1} \log \mathbb{E}_{x \sim Q} \left(\frac{P(x)}{Q(x)}\right)^{\alpha} \tag{3}$$

*then, a randomized mechanism* $\mathcal{M}$ : $\mathcal{X}$ $\rightarrow$ $\mathcal{R}$ *is* $(\alpha, \epsilon)$ *Rényi differential privacy, if for any two adjacent datasets* $\mathcal{D}$, $\mathcal{D}'$ *and for any measurable sets* $\mathcal{S}$:

$$R_{\alpha}(\mathcal{M}(D)||\mathcal{M}(D')) \le \epsilon \tag{4}$$

**Theorem 1** *Mironov (2017) showed that if* $\mathcal{M}$ *is* $(\alpha, \epsilon)$-*RDP guarantee, is also* $(\epsilon + \frac{\log 1/\delta}{\alpha - 1})$-*DP.*

In this section, we provide more detailed explanation of privacy preserving in PRISM and also present updated results when DP is applied. To satisfy the $(\epsilon, \delta)$-DP, our goal is privatize the probability vector $\theta \in [0, 1]^d$ by adding gaussian noise $\mathcal{N}(0, \sigma^2)$, where $\sigma^2 = \frac{2 ln(1.25/\delta)\Delta_2^2}{\epsilon^2}$ and $\Delta_2 = \max_{D, D'} ||\mathcal{M}(D) - \mathcal{M}(D')||_2$. When the local training is end, each client has scores $s \in \mathbb{R}^d$ to choose which weight to prune. Recall that probaility $\theta \in [0, 1]^d$ can be obtained through sigmoid function. We inject gaussian noise and clip to $\tilde{\theta} \in [c, 1 - c]^d$, where c is a small value $0 < c < \frac{1}{2}$. In out setup, we fix it at 0.1. Now, we ensure $\tilde{\theta}$ is $(\epsilon, \delta)$-DP. For a fair comparison, we use $(\epsilon, \delta) = (50, 10^{-5})$ to PRISM and our baselines in all of our experiments. In addition, we regulate the global round to ensure that the overall privacy budget does not exceed $\epsilon$. To track the overall privacy budget, we employ subsampled moments accountant Wang et al. (2019). We refer to the Opacus library which is the user-friendly pytorch framework for differential privacy Yousefpour et al. (2021).

Imola & Chaudhuri (2021); Isik et al. (2022) have shown that performing post processing to already privatized vector $\tilde{\theta}$ such as bernoulli sampling enjoys privacy amplification under some conditions. By doing so, the overall privacy budget becomes smaller $\epsilon_{amp} \le \min\{\epsilon, d\gamma_{\alpha}(c)\}$, where $\gamma_{\alpha}(\cdot)$ is the binary symmetry Rényi divergence as expressed below:

$$\gamma_{\alpha}(c) = \frac{1}{\alpha - 1} \log(c^{\alpha}(1 - c)^{\alpha} + (1 - c)^{\alpha}c^{1-\alpha}), \tag{5}$$

where $\alpha$ refers to the order of the divergence. Note that $d$ limits the privacy amplification when the model size becomes large. Since PRISM assumes that the model size is large enough due to SLT, we focus on communication efficiency rather than privacy amplification.

# B    TRAINING DETAILS

In this section, we provide the detailed description of our implementations and experimental settings. Our code is based on Santos et al. (2019); Yeo et al. (2023). They employ the ImageNet-pretrained VGG19 network for feature matching by minimizing the Eq. 1. However, calculating the first and second moments require the large batch size to obtain the accurate statistics. To address this issue, Santos et al. (2019) introduces Adam moving average (AMA). With a rate $\lambda$, the update of AMA $m$ is expressed as follows:

$$m \leftarrow m - \lambda \text{ADAM}(m - \Delta), \tag{6}$$

Table 4: **Quantitative comparison in IID scenario.** We compare the FID, P&R, D&C, communication cost with $(\epsilon, \delta) = (50, 10^{-5})$. Communication cost is reported by capturing the number of bytes exchanged from the client to the server. We set $\alpha = 70$ for PRISM-$\alpha$ while $\alpha = 100$ for PRISM-$\alpha$ (full score). Since MD-GAN and PRISM-$\alpha$ (full score) do not provide privacy protection, they are considered as upper bounds.

| Method (comm.cost) | Privacy | Metric | MNIST | CelebA |
|---|---|---|---|---|
| MD-GAN (14MB) | ✗ | FID ↓ | 5.5364 | 6.74 |
| | | P&R ↑ | 0.8081 / 0.7414 | 0.796 / 0.6243 |
| | | D&C ↑ | 0.7525 / 0.7955 | 1.05 / 0.8919 |
| GS-WGAN (15MB) | ✓ | FID ↓ | 68.53 | 203.6972 |
| | | P&R ↑ | 0.0975 / 0.0794 | 0.1675 / 0 |
| | | D&C ↑ | 0.0257 / 0.0367 | 0.0357 / 0.002 |
| DP-FedAvgGAN (14MB) | ✓ | FID ↓ | 87.9032 | 206.0260 |
| | | P&R ↑ | 0.2237 / 0.0688 | 0.0518 / 0.0585 |
| | | D&C ↑ | 0.0549 / 0.0225 | 0.0111 / 0.0017 |
| PRISM (5.75MB) | ✓ | FID ↓ | 40.8520 | 52.9511 |
| | | P&R ↑ | 0.6043 / 0.11 | 0.6154 / **0.1171** |
| | | D&C ↑ | 0.4074 / 0.295 | **0.3781** / 0.2268 |
| PRISM-70 (9.6MB) | ✓ | FID ↓ | 31.4345 | **40.7070** |
| | | P&R ↑ | **0.4263** / 0.5051 | **0.6169** / 0.0978 |
| | | D&C ↑ | **0.1624** / 0.1786 | 0.378 / 0.268 |
| PRISM-100 (full score) (24MB) | ✓ | FID ↓ | **31.1024** | 41.1423 |
| | | P&R ↑ | 0.4051 / **0.5127** | 0.5884 / 0.1063 |
| | | D&C ↑ | 0.1535 / **0.3549** | 0.2618 / **0.3719** |

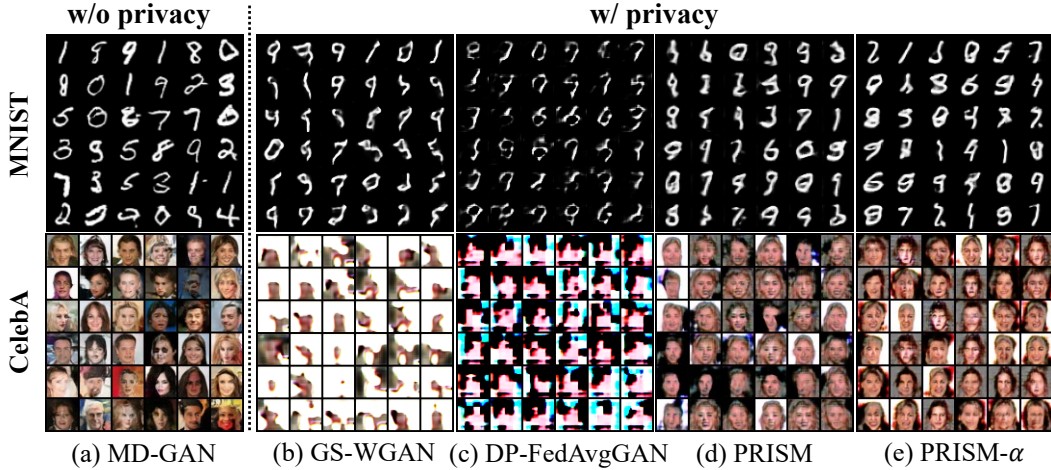

**w/o privacy**      **w/ privacy**

MNIST    CelebA

(a) MD-GAN    (b) GS-WGAN   (c) DP-FedAvgGAN    (d) PRISM      (e) PRISM-$\alpha$

Figure 8: **Qualitative results in IID scenario.** We compare the generated images from the models in Table 1 on MNIST, CelebA, and CIFAR10 with $(\epsilon, \delta) = (50, 10^{-5})$. Here, $\alpha = 70$ for PRISM-$\alpha$.

where ADAM denotes Adam optimizer Kingma & Ba (2014) and $\Delta$ is the discrepancy of the means of the extracted features. Note that ADAM$(m - \Delta)$ can be interpreted as gradient descent by minimizing the L2 loss:

$$\min_m \frac{1}{2} \|m - \Delta\|^2. \tag{7}$$

This means the difference of statistics $(m - \Delta)$ is passed through a single MLP layer and updated using the Adam optimizer to the direction of minimizing Eq. 7. By utilizing AMA, Eq. 1 is formulated as $\mathcal{L}_{MMD}^k = \left\| \mathbb{E}_{x \sim \mathcal{D}^k}[\psi(x)] - \mathbb{E}_{y \sim \mathcal{D}_{fake}^k}[\psi(y)] \right\|^2 + \left\| \text{Cov}(\psi(\mathcal{D}^k)) - \text{Cov}(\psi(\mathcal{D}_{fake}^k)) \right\|^2$, Algorithm 1, 2 provides the psuedocode for PRISM and PRISM-$\alpha$ correspondingly. AMA is omitted to simply express the flow of PRISM. See our code for pytorch implementation. We train the local generator for 100 local iterations with learning rate of 0.1. For the AMA layer, learning rate is

Table 5: **Quantitative comparison in non-IID scenario.** We compare the FID, P&R, D&C, and communication cost with $(\epsilon, \delta) = (50, 10^{-5})$. We report the number of bytes exchanged from the client to the server as communication cost. As in Table 4, we set $\alpha = 70$ for PRISM-$\alpha$, while $\alpha = 100$ for PRISM-$\alpha$ (full score). MD-GAN and PRISM-$\alpha$ (full score) do not preserve privacy.

| Method (comm.cost) | privacy | Metric | MNIST | CelebA |
|---|---|---|---|---|
| MD-GAN (14MB) | ✗ | FID ↓ | 40.4037 | 18.5903 |
| | | P&R ↑ | 0.3297 / 0.491 | 0.715 / 0.4673 |
| | | D&C ↑ | 0.1145 / 0.1434 | 0.7363 / 0.6793 |
| GS-WGAN (15MB) | ✓ | FID ↓ | 155.2829 | 246.682 |
| | | P&R ↑ | 0.0216 / 0.0011 | 0.2318 / 0.0 |
| | | D&C ↑ | 0.0044 / 0.0009 | 0.0566 / 0.0015 |
| DP-FedAvgGAN (14MB) | ✓ | FID ↓ | 131.6076 | 216.2886 |
| | | P&R ↑ | 0.3086 / 0.0608 | 0.0639 / 0.0 |
| | | D&C ↑ | 0.0674 / 0.0113 | 0.0140 / 0.0015 |
| PRISM (5.75MB) | ✓ | FID ↓ | 50.7144 | 57.3357 |
| | | P&R ↑ | 0.3666 / 0.3277 | 0.5063 / 0.0712 |
| | | D&C ↑ | 0.1308 / 0.1035 | 0.2536 / 0.1499 |
| PRISM-70 (9.6MB) | ✓ | FID ↓ | 43.8706 | 44.6085 |
| | | P&R ↑ | 0.3769 / 0.3878 | **0.5916 / 0.0817** |
| | | D&C ↑ | 0.1348 / 0.131 | 0.3289 / **0.2422** |
| PRISM-100 (full score) (24MB) | ✓ | FID ↓ | **38.7667** | **43.3729** |
| | | P&R ↑ | **0.3825 / 0.3919** | 0.5814 / 0.0772 |
| | | D&C ↑ | **0.1457 / 0.1457** | **0.3519** / 0.2317 |

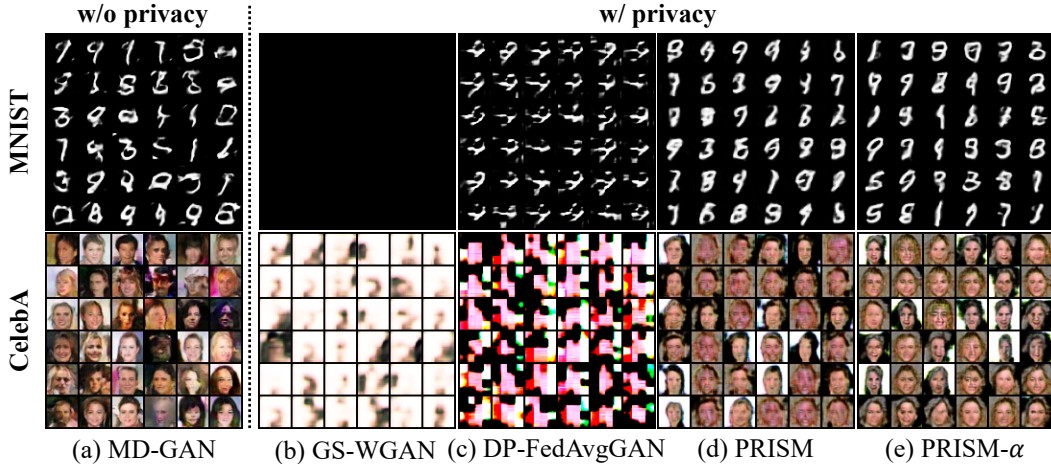

(a) MD-GAN  (b) GS-WGAN  (c) DP-FedAvgGAN  (d) PRISM  (e) PRISM-$\alpha$

Figure 9: **Qualitative results in Non IID scenario.** We compare the generated images from the models in Table 2 on MNIST, CelebA, and CIFAR10 with $(\epsilon, \delta) = (50, 10^{-5})$. Here, $\alpha = 70$ for PRISM-$\alpha$.

set to 0.005. In addition, we use the Adam optimizer with $\beta_1 = 0.5, \beta_2 = 0.999$ to update the scores of the generators. After all clients complete their training, communication round is initiated. We set the global epoch to 150 for the MNIST dataset and 350 for the CelebA and CIFAR10 datasets. As we do not adjust the parameters, note that there is room for performance improvements through hyperparameter tuning.

## C ADDITIONAL EXPERIMENTS

In this section, we provide additional experiments to validate the effectiveness of MMD loss and solve the potential two questions. One can ask why use MMD? To answer this, we conduct an

Table 6: **Quantitative comparison in IID scenario.** We compare the FID, P&R, D&C, communication cost with $(\epsilon, \delta) = (9.8, 10^{-5})$. Communication cost is reported by capturing the number of bytes exchanged from the client to the server. We set $\alpha = 70$ for PRISM-$\alpha$ while $\alpha = 100$ for PRISM-$\alpha$ (full score).

| Method (comm.cost) | Privacy | Metric | MNIST | CelebA |
|---|---|---|---|---|
| MD-GAN (14MB) | ✗ | FID ↓ | 5.5364 | 6.74 |
| | | P&R ↑ | 0.8081 / 0.7414 | 0.796 / 0.6243 |
| | | D&C ↑ | 0.7525 / 0.7955 | 1.05 / 0.8919 |
| GS-WGAN (15MB) | ✓ | FID ↓ | 328.3735 | 338.6659 |
| | | P&R ↑ | 0.0 / 0.0 | 0.0 / 0.0 |
| | | D&C ↑ | 0.0 / 0.0 | 0.0 / 0.0 |
| DP-FedAvgGAN (14MB) | ✓ | FID ↓ | 149.3563 | 243.2313 |
| | | P&R ↑ | 0.0193 / 0.0 | 0.1008 / 0.0 |
| | | D&C ↑ | 0.0041 / 0.0013 | 0.0211 / 0.0013 |
| PRISM (5.75MB) | ✓ | FID ↓ | 59.4503 | 101.2505 |
| | | P&R ↑ | 0.2594 / 0.3884 | 0.4306 / **0.1051** |
| | | D&C ↑ | 0.0832 / 0.0852 | 0.1794 / 0.0889 |
| PRISM-70 (9.6MB) | ✓ | FID ↓ | **45.652** | 90.413 |
| | | P&R ↑ | **0.3876** / 0.3523 | **0.4307** / 0.0786 |
| | | D&C ↑ | **0.1382** / **0.1087** | **0.1834** / 0.0989 |
| PRISM-100 (full score) (24MB) | ✓ | FID ↓ | 50.441 | **74.4931** |
| | | P&R ↑ | 0.3307 / **0.3947** | 0.4225 / 0.0904 |
| | | D&C ↑ | 0.1093 / 0.0923 | 0.1794 / **0.1069** |

Table 7: **Quantitative comparison in non-IID scenario.** We compare the FID, P&R, D&C, and communication cost with $(\epsilon, \delta) = (9.8, 10^{-5})$. We report the number of bytes exchanged from the client to the server as communication cost.

| Method (comm.cost) | privacy | Metric | MNIST | CelebA |
|---|---|---|---|---|
| MD-GAN (14MB) | ✗ | FID ↓ | 5.5364 | 6.74 |
| | | P&R ↑ | 0.8081 / 0.7414 | 0.796 / 0.6243 |
| | | D&C ↑ | 0.7525 / 0.7955 | 1.05 / 0.8919 |
| GS-WGAN (15MB) | ✓ | FID ↓ | 338.6659 | 338.4467 |
| | | P&R ↑ | 0.0 / 0.0 | 0.0 / 0.0 |
| | | D&C ↑ | 0.0 / 0.0 | 0.0 / 0.0 |
| DP-FedAvgGAN (14MB) | ✓ | FID ↓ | 146.0803 | 266.3443 |
| | | P&R ↑ | 0.0779 / 0.0002 | 0.0187 / 0.0 |
| | | D&C ↑ | 0.0207 / 0.0026 | 0.0034 / 0.0003 |
| PRISM (5.75MB) | ✓ | FID ↓ | 81.0177 | 91.2942 |
| | | P&R ↑ | 0.3391 / **0.2996** | 0.3836 / 0.0801 |
| | | D&C ↑ | 0.107 / 0.0497 | 0.1531 / 0.0791 |
| PRISM-70 (9.6MB) | ✓ | FID ↓ | 74.4354 | 84.1179 |
| | | P&R ↑ | 0.2967 / 0.2534 | **0.4322** / 0.0795 |
| | | D&C ↑ | 0.0923 / 0.0573 | 0.1756 / **0.1** |
| PRISM-100 (full score) (24MB) | ✓ | FID ↓ | **62.4888** | **80.257** |
| | | P&R ↑ | **0.3554** / 0.2959 | 0.4123 / **0.081** |
| | | D&C ↑ | **0.1226** / **0.0961** | **0.1778** / 0.0993 |

experiment which employs GAN loss instead of MMD loss. In this setup, we train both generator and discriminator at client side and aggregate only generator likewise PRISM. Figure 10 shows that application of GANloss to PRISM SLT actually does not work. Second possible question is that does PRISM need multiple round? Since PRISM finds the strong lottery ticket in the dense network, someone may be curious about the reason for performing multiple rounds rather than one

---

**Algorithm 1** PRISM

---

**Parameter:** learning rate $\eta$, communication rounds $T$, local iterations $I$
**Input:** local datasets $\cup_{k=1}^{K}\mathcal{D}^k$, ImageNet pretrained VGGNet $\psi$, random noise $z$

    **Server execute:**
    Initialize a random weight $W_{init}$ and score vector $s$, then broadcasts to all clients.
    **for** *round t = 1, ..., T* **do**
        **Client side:**
        **for** *each client $k \in [1, K]$* **do**
            $s_t^k = s_t$                                       ▷ Download score vector
            **for** *local iteration $i = 1, , , L$* **do**
                $\theta_t^k \leftarrow \text{Sigmoid}(s_t^k)$
                $M_t^k \sim \text{Bern}(\theta_t^k)$
                $W_t^k \leftarrow W_{init} \odot M_t^k$
                $\mathcal{D}_{fake}^k \leftarrow W_t^k(z)$                      ▷ Generate fake images
                Extract real and fake features $\psi(\mathcal{D}^k), \psi(\mathcal{D}_{fake}^k)$
                $s_t^k \leftarrow s_t^k - \eta\nabla\mathcal{L}_{MMD}^k(\psi(\mathcal{D}^k), \psi(\mathcal{D}_{fake}^k))$    ▷ Update local score vector
            **end for**
            $\bar{\theta_t^k} \leftarrow \text{Sigmoid}(s_t^k)$
            $\tilde{\theta_t^k} =\leftarrow \bar{\theta_t^k} + \mathcal{N}(0, I\sigma^2)$
            Clip to [c, 1-c]
            $M_t^k \sim \text{Bern}(\theta_t^k))$
            Upload binary mask $M_t^k$ to the server.
        **end for**
        **Server side:**
        $\theta_{t+1} \leftarrow \sum_{k=1}^{K} M_t^k$                     ▷ Aggregate the received binary masks
        $s_{t+1} \leftarrow \text{Sigmoid}^{-1}(\theta_{t+1})$
    **end for**
    Sample the supermask $M^* \sim \text{Bern}(\theta_T)$
    Obtain the final model $W^* \leftarrow W_{init} \odot M^*$

---

**Algorithm 2** PRISM-$\alpha$

---

**Input:** ratio of score layer $\alpha$
**Output:** probability $\theta_t^k(100 - \alpha)$ and binary mask $M_t^k(\alpha)$

    **Client side:**
    **for** *layer $l = 1, ..., L$* **do**
        **if** IsScoreLayer(l,$\alpha$,L) **then**
            Return probability $\theta_t^k(l)$
        **else**
            Return binary mask $M_t^k(l)$
        **end if**
    **end for**

---

shot manner. To address this question, we draw an analogy between each local dataset in our method and a mini-batch in traditional centralized SGD. In this comparison, one-shot aggregation is akin to optimizing with a very large step size. It is well-known in optimization theory that training with a large step size and few epochs can often lead to local minima. Coming back to the perspective of federated learning, by aggregating the scores (or binary masks) of each client through multiple rounds, our goal is to find the global lottery ticket that is not biased towards specific local dataset. To support our opinion, we conduct an experiment in which we performed single aggregation after overfitting MNIST dataset to 500 local iterations. The results can be found in Figure 11. We observe that it fails to generate proper images

Another interesting point of view is the effect of the choice of architecture on the performance of PRISM. In the previous experiments, we have consistently utilized the ResNet-based generator for

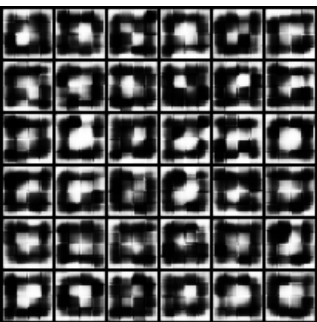

Figure 10: Generated images of PRISM + GAN loss.

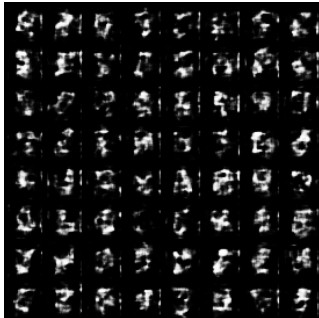

Figure 11: Generated images of PRISM with one shot aggregation.

both our method and all baseline methods to ensure a fair comparison. In principle, our method should work generally well for different architectures. Basically, SLT just presumes that a dense network is sufficiently overparameterized to find a useful subnetwork. This implies that SLT has robustness with architecture. To validate our argument, we evaluate architectures frequently used in generative models in Table 8 while keep all setting, including DP, consistent with our framework. Our default configuration is a ResNet-based generator, and architectures such as DCGAN and SNGAN show robust performance on MNIST but exhibit a slight advantage depending on the architecture in CelebA. PRISM can somewhat follow the performance of the architecture since it extracts a subnetwork from the dense network. Still, it consistently finds a stable SLT for various structures.

Table 8: **The effect of the choice of architecture.** Left table shows the results in MNIST dataset, while right one demonstrates on the CelebA datset.

| MNIST | | | | CelebA | | | |
|---|---|---|---|---|---|---|---|
| **Metric** | **ResNet** | **DCGAN** | **SNGAN** | **Metric** | **ResNet** | **DCGAN** | **SNGAN** |
| FID | 34.4595 | 36.9683 | 33.4447 | FID | 51.0857 | 78.4130 | 38.3009 |
| Precision | 0.3784 | 0.3694 | 0.4652 | Precision | 0.5267 | 0.4540 | 0.6851 |
| Recall | 0.4213 | 0.4848 | 0.3318 | Recall | 0.0631 | 0.0404 | 0.0925 |
| Density | 0.1418 | 0.1323 | 0.2047 | Density | 0.2676 | 0.1916 | 0.5378 |
| Coverage | 0.1588 | 0.1490 | 0.2019 | Coverage | 0.2012 | 0.0947 | 0.3155 |

