# OpenReview forum: "PRISM: Privacy-Preserving Improved Stochastic Masking For Federated Generative Models"
_ICLR.cc/2024/Conference — Submitted to ICLR 2024_

### Official Review · Reviewer_xZ6u · 2023-10-31

**Soundness:** 1 poor
**Presentation:** 2 fair
**Contribution:** 1 poor
**Rating:** 3
**Confidence:** 3

**Summary:**

The paper proposes a certain binary masking technique for distributed training of generative models. Instead of communicating the model weights, only binary masks are communicated in a stochastic manner. Using the lottery ticket hypothesis, and by combining the masks, aggregator can obtain an efficient generative model.

**Strengths:**

- The paper seems well written and is easy to read.

- The idea of sending only the masks seems interesting. As far as I see, this paper considers a distributed version of the methods proposed by Isak et al. (2022) and Li et al. (2021a).

**Weaknesses:**

One clear deficit of the paper is that DP aspects of the method are not discussed although DP is heavily emphasised in the title, abstract and intro ("privacy-preserving" in the title, differential privacy explicitly mentioned elsewhere). I can see DP discussed only in those seven lines of Section 3.2.

There is no DP analysis for the method and no $\varepsilon$'s or $\delta$'s are reported in the experimental results.

If I understand the method correctly, the masks would indeed depend on the data, so the $\varepsilon$ is definitely not $0$.

**Questions:**

- How would you prove the differential privacy guarantees for the proposed method?

- What would be the DP guarantees for the experimental results that you provide?

---

> ### Author Response · Authors · 2023-11-19
> **[Response 1/1] Thank for your constructive feedbacks!**
>
> **R5-1. Differential privacy**
>
> Please see GC1.

---

> > ### Comment · Reviewer_xZ6u · 2023-11-20
> >
> > I think the epsilon-value 50 is very large and not really in line with what is commonly used.
> >
> > This makes me wonder whether DP is suitable for this method:
> >
> > > Since PRISM assumes that the model size is large enough (due to SLT), we focus on communication efficiency rather than privacy amplification due to bernoulli sampling.

---

> > > ### Author Response · Authors · 2023-11-21
> > > **Thanks for reply!**
> > >
> > > Thank you for your prompt feedback. We initially set $\epsilon=50$ and $\delta=0.00001$ to align with the baseline parameters used in their original configurations, ensuring a fair comparison. However, in response to your concern about the high epsilon-value, we conducted an additional experiment with a lower privacy cost of $\epsilon=9.8$, commonly adopted in Differential Privacy Stochastic Gradient Descent (DP-SGD). As detailed in Tables 6 and 7, PRISM consistently demonstrated robust performance, even under stricter privacy constraints. In contrast, both DP-FedAvgGAN and GS-WGAN showed performance collapse. These models necessitate higher $\epsilon$ values, as evidenced in Table 2 of the GS-WGAN study [1].
> > >
> > > [1] Chen, Dingfan, Tribhuvanesh Orekondy, and Mario Fritz. "Gs-wgan: A gradient-sanitized approach for learning differentially private generators." Advances in Neural Information Processing Systems 33 (2020): 12673-12684.

---

> > > > ### Comment · Reviewer_xZ6u · 2023-11-22
> > > >
> > > > Thank you for the reply. I still cannot really follow the privacy analysis, to me it seems there are only the Rényi DP parameters of the Bernoulli sampling used. In the RDP analysis, the Gaussian mechanism RDP parameters do not seem to show up. If those are not needed, then why would one perturb that $\theta$-vector.

---

> > > > > ### Comment · Reviewer_xZ6u · 2023-11-22
> > > > >
> > > > > Ps. the paper seems to be currently way over 9 pages.

---

> ### Author Response · Authors · 2023-11-23
> **Response to Reviewer xZ6u:**
>
> Thanks for providing valuable feedbacks and various concerns that arise in our work.
>
> As can be seen from Appendix A of our manuscript, after clipping the noise-injected $\theta$, the privacy budget we have becomes $\epsilon_{amp}\leq min\{\epsilon, d\gamma_{\alpha}(c)\}$. Here, during the noise injection process, the variance $\sigma^2$ of the Gaussian noise should become a function of $\epsilon$, which controls the level of privacy we guarantee. Overall, the perturbing $\theta$ is necessary to guarantee a certain level of privacy.
> Regarding the privacy amplification, our intention was to theoretically show that the privacy bound can become tighter (or at least does not compromise the bound) by taking advantage of Bernoulli sampling. Depending on various parameters and the model size, it may or may not be amplified, but the key point here is that since we have applied $(\epsilon, \delta)$-dp to $\theta$, there is no case that the privacy bound falls below $\epsilon$.

---

> ### Comment · Reviewer_xZ6u · 2023-11-23
>
> Thank you for the reply. This is the so-called post-processing property of DP: no matter what data-independent function you apply on the DP output (stochastic or not), the epsilons will not increase (they can decrease though). Thus your privacy guarantees will not get worse with the Bernoulli sampling procedure. But I can imagine that the Bernoulli sampling part could even amplify the privacy guarantees. But that should be then shown mathematically, how much amplification happens. And in case you are not actually using the amplification, as I understand, that should also be clearly stated and also the used privacy bounds should be clearly stated (if they are, e.g., the RDP-bounds of the Gaussian mechanism).

---

> > ### Author Response · Authors · 2023-11-23
> > **Response to Reviewer xZ6u**
> >
> > As the reviewer pointed out, bernoulli sampling allows privacy amplifications when  $d\gamma_{\alpha}(c)<\epsilon$ holds, according to the bound $\epsilon _{amp}<\leq \min$ {$ \epsilon, d\gamma _{\alpha}(c) $}.
> >
> >  In our setup where the target epsilon is 9.8 and c=0.1, privacy amplification can be achieved when $d \leq 2450$ because the $RDP_{\alpha}(c)$ value per global round is 0.004, where $\alpha=2.8$.

---

### Official Review · Reviewer_a69J · 2023-11-01

**Soundness:** 3 good
**Presentation:** 3 good
**Contribution:** 1 poor
**Rating:** 5
**Confidence:** 4

**Summary:**

This paper proposes combining the edge-popup algorithm, initially proposed for learning subnetworks in randomly initialized networks, with the problem of training generative models in the federated setting.  During each FL round, clients send a binary mask sampled from the weight scores, instead of the scores themselves, which can save on communication cost. The final learned mask can be applied on the randomly initialized model to obtain an efficient model for inference, since weights are initialized (and kept frozen) from {-sigma, sigma}.

**Strengths:**

- The paper tackles an important subject: efficient and private federated training of generative models
- The paper is well presented, and cites the relevant literature
- The proposed method is simple, and the experimental results back the authors' claims
- The authors experiment with wide variety of datasets, and compare with multiple baselines

**Weaknesses:**

My main issue with this paper is the lack of novelty. It is a direct application of two previous works:
- Sparse Random Networks for Communication-Efficient Federated Learning (https://arxiv.org/pdf/2209.15328.pdf), which applies the exact same masking and sampling scheme proposed in this paper, in a federated setting, for classification tasks, even the privacy preserving aspect of the proposed paper is inherited from this paper, as mention in S4.3
- Can We Find Strong Lottery Tickets in Generative Models? (https://arxiv.org/pdf/2212.08311.pdf), which applies the same subnetwork strategy (edge-pop) for training GANs, and proposes to use the MDD loss for stable training

This paper just simply combines the two methods to get an FL subnetwork method for generative models, and offers no extra insight beyond those two works in my opinion. It is unfortunately below the acceptance threshold for ICLR, which is why I do not recommend acceptance.




Misc:
- In Sec 3.1, the edge-pop algorithm was proposed in (https://arxiv.org/pdf/1911.13299v2.pdf), and not in (https://arxiv.org/pdf/1803.03635.pdf)
- In Tables 1 and 2, PRISM-alpha entries should be directly replaced with the alpha value, since it is fixed, e.g. PRISM-100 and PRISM-70. It is very confusing otherwise.
- it appears that GANs are growing slightly out of fashion in the generative AI community, which slightly affects the significance of this work.

**Questions:**

- Did the authors experiment with other losses, to further showcase the strength of MDD? It would provide more insight to the results of Tables 1 and 2 if the authors applied the same losses form other methods with their subnetwork strategy. This way one can better understand where the performance improvements are coming from.

- In Tables 1 and 2, what does it mean when an algorithm is private vs. non-private? this seems very reductive and grossly simplifies the notion of privacy. It would be better to quantify it and report that instead. For instance, the hybrid method PRISM-alpha remains "private" for all alpha <100, and suddenly becomes non-private when alpha=100?

---

> ### Author Response · Authors · 2023-11-19
> **[Response 1/1] Thank for your constructive feedbacks!**
>
> **R4-1. Experiment with other losses.**
>
> Please see GC1 and R1-1.
> We want to clarify the distinction between our approach and baselines. Baselines typically  train the weights using GAN loss, while PRISM employs the MMD loss to find the SLT within the dense network. If an SLT is introduced into a baselines method, it essentially becomes PRISM. In that case, there is a potential question to consider whether strong performance emerges when GAN loss is applied to PRISM. To handle this curiosity, we conducted an experiment which utilizes GAN loss as score function for SLT, not MMD loss. In order to learn the generator by GAN loss instead of MMD loss, each client learns the discriminator and generator adversarially. After the local training is completed, the server aggregates and returns only the generator. As seen in Figure.10, the unstable GAN loss fails to find the SLT and proceed the FL normally, which implies that MMD loss is suitable for decentralized setting and SLT.
>
> **R4-2. Differential privacy**
>
> Please see GC1.
>
> **R4-3. Limited novelty**
>
> Please see GC2.
>
> **R4-4. GAN is out of fashion**
>
>  The tackiness of GAN is discussed in GC1, GC2. Additionally, the reviewer’s question seems to be intended for DDPM, which has recently shown strong performance. Although we have been empirically confirmed that SLT works successfully, there are still several limitations such as slow convergence.
> We would like to clarify that PRISM uses MMD loss, while the baselines employ GANs. As you mentioned, it’s true that the popularity of GANs has declined with the emergence of diffusion models, but GANs still offer valid advantages such as simplicity and fast inference times [1]. While it’s possible to apply diffusion models to FL setups, the introduction is not straightforward in FL, where communication efficiency becomes a bottleneck. This is because diffusion models require substantial computational resources and slow convergence due to the iterative learning. Importantly, there should be discussions considering various ways in the field of FL for generative models.
>
> **R4-5. Other minor comments in Misc.**
>
> Corrected.
>
> [1] Kang, Minguk, et al. "Scaling up gans for text-to-image synthesis." Proceedings of the IEEE/CVF Conference on Computer Vision and Pattern Recognition. 2023.

---

> > ### Author Response · Authors · 2023-11-23
> > **Update the information in GC1**
> >
> > We provide additional content updates conducted with $\epsilon=9.8$ on GC1. Please refer to Table.6 and 7 in our manuscript.

---

### Official Review · Reviewer_sDDa · 2023-11-01

**Soundness:** 3 good
**Presentation:** 2 fair
**Contribution:** 2 fair
**Rating:** 5
**Confidence:** 3

**Summary:**

This paper proposes a new federated framework tailored for generative models named PRISM, which emphasizes not only strong and stable performance but also resource efficiency and privacy preservation. Experimental results on MNIST, CelebA, and CIFAR10 demonstrate that PRISM outperforms the previous methods in both IID and non-IID cases, all while preserving privacy at the lowest communication cost.

**Strengths:**

1. The investigated problem is novel, which addresses the challenges on communication efficiency, privacy, and performance instability altogether for federated generative models.

2. PRISM exhibits robust performance even in non-IID FL settings, unlike traditional GAN-based approaches.

**Weaknesses:**

1. The existing methods have not been comprehensively analyzed for their challenges.  For example, the proposed PRISM is designed with the objective of surpassing existing methods in terms of stable performance, resource efficiency, and privacy preservation. However, the introduction lacks an analysis of the challenges related to privacy preservation in existing works.

2. Is this approach outperforming compared to traditional FL? On one hand, can PRISM attain equivalent efficiency to traditional FL, which directly uploads local parameters? On the other hand, when we consider the attack of inferring the client's local dataset, can PRISM provide robust protection against such attacks? We look forward to the authors conducting relevant experiments to address the questions.

3. How can DP guarantee privacy in generative models? Any proofs?

4. The comparison experiments in the study may not reflect the state-of-the-art, and we anticipate that PRISM will be compared with more advanced methods. For example, whether PRISM's model performance surpasses that of Multi-FLGAN.

**Questions:**

see weaknesses

---

> ### Author Response · Authors · 2023-11-19
> **[Response 1/1] Thank for your constructive feedbacks!**
>
> **R3-1. Differential privacy**
>
> Please see GC1.
>
> **R3-2. Comparison to traditional FL**
>
> Since PRISM uploads binary masks instead of float-type weights, PRISM can save communication costs up to 32 times compared to traditional FL. The comparison of efficiency between PRISM and traditional FL can be found in Table.1,2, and 3.
> On the aspect of security, particularly against attacks aiming to infer clients' local datasets, PRISM offers robust protection. This security is primarily attributed to three factors. First, even if an attacker hijacks the mask used in PRISM, without knowledge of the client's architecture, the information gleaned is effectively meaningless. Second, even if the attacker is aware of the architecture, the lack of precise information about the random initialization renders any attempt to decode the client's data futile. Third, PRISM incorporates differential privacy (DP) techniques. It is well known that DP can prevent various attacks in decentralized environments. For more details on DP, please see GC1 and R3-3.
>
> **R3-3. DP in generative model**
>
> Differential privacy is an algorithm designed to reduce the influence of single elements on output. Let’s suppose M is a mechanism that we want to guarantee privacy, such as neural network. With adjacent datasets D, D’, M is $(\epsilon, \delta)$-differential privacy when $P(M(D)) \leq e^{\epsilon}P(M(D’)) + \delta$ is satisfied. Also, by composability, if all components of M satisfy DP, M itself also satisfies DP. Since generative model is a stacked layer, DP guarantees privacy in generative models.
>
> **R3-4. Comparison with SOTA (e.g., Multi-FLGAN).**
>
> Multi-FLGAN introduces multi-GAN, competing X generators and Y discriminators to select the best-performing generator. Consequently, there exist X*Y sync servers. While Multi-FLGAN achieves excellent performance and stability, it demands a substantial amount of computation resources. According to the paper, authors used four V100 GPUs for experiments on MNIST and FMNIST. However, we agree that comparing PRISM with SOTA models can enhance the contribution of PRISM. We will apply differential privacy to Multi-FLGAN and update the results when experiments are completed.
>
> **(updated)**
>
> As we discussed above, Multi-FLGAN employs a complex architecture involving multiple generators and discriminators ($X\times Y$ of each per client), with the server aggregating $N\times X\times Y$ models, where $N$ is the number of clients. Our comparison between PRISM and Multi-FLGAN[1] involved a scaled-down Multi-FLGAN setup ($X=2, Y=2$) due to its high resource demands.
> The table below details our findings, comparing the standard Multi-FLGAN (default setup introduced in the original paper) and an modified version, Multi-FLGAN+DP version, which incorporates differential privacy ($(\epsilon, \delta)=(9.8, 10^{-5})$) for a fair comparison to PRISM and other baseline models. We observed significant performance degradation in Multi-FLGAN when applying DP, likely due to its complex framework involving both generators and discriminators.
> This raises to two key concerns:
> DP suitability: The complex framework of Multi-FLGAN limits its effectiveness when DP is applied, suggesting that the framework may not be ideally suited for privacy-preserving FL scenarios.
> Resource and Communication Overheads: The architecture demands extensive training resources and incurs high communication overhead, posing challenges for practical deployment, especially in resource-constrained environments.
>
>
> |        MNIST              |   Privacy     |      FID      |      P&R                  |      D&C                 |     Cost     |
> |:---------------------------:|:----------------:|:---------------:|:--------------------------:|:-------------------------:|:---------------:|
> |  Multi-FLGAN          |        X        |   44.97      |   0.1461 / 0.3248   |   0.0461 / 0.0868   |    52MB    |
> |  Multi-FLGAN + DP |        O        |  323.721   |         0.0 / 0.0         |         0.0 / 0.0         |    52MB    |
> |  PRISM                    |        O        |  59.4503   |   0.2594 / 0.3884   |   0.0832 / 0.0852   |   5.75MB  |
>
> [1] Amalan, Akash, et al. "MULTI-FLGANs: Multi-Distributed Adversarial Networks for Non-IID distribution." arXiv preprint arXiv:2206.12178 (2022).

---

> > ### Comment · Reviewer_sDDa · 2023-11-21
> > **Reply to the Authors' Response**
> >
> > Thanks for your response! I am still willing to see the comparison results. Also, as pointed out by Reviewer xZ6u, epsilon=50 seems pretty large, making me doubt the practicality.

---

> > > ### Author Response · Authors · 2023-11-21
> > > **Thanks for reply!**
> > >
> > > Thank you for your prompt response and sharing valuable insights. We are continuously working to provide a comparison with Multi-FLGAN before the upcoming deadline. Your second comment raises concerns about the confidence in the high $\epsilon$. Firstly, we would like to note that, for a fair comparison in additional experiments presented in GC1, both the baselines and PRISM used $\epsilon=50$. As for your concern regarding the commonly used $\epsilon=9.8$ in DP-SGD, we conducted additional experiments, and reviewers can find the results in Table.6,7. Despite the tighter privacy cost, PRISM showed a somewhat acceptable performance drop, while DP-FedAvgGAN and GS-WGAN exhibited collapse. Also, as indicated in Table.2 of GS-WGAN[1], it is evident that both DP-FedAvgGAN and GS-WGAN used extremely high $\epsilon$.
> > >
> > > [1] Chen, Dingfan, Tribhuvanesh Orekondy, and Mario Fritz. "Gs-wgan: A gradient-sanitized approach for learning differentially private generators." Advances in Neural Information Processing Systems 33 (2020): 12673-12684.

---

> ### Author Response · Authors · 2023-11-22
> **Response for R3-4 is updated**
>
> Our response about the comparisons with other SOTA (e.g., Multi-FLGAN) is updated in R3-4.

---

### Official Review · Reviewer_pni9 · 2023-11-01

**Soundness:** 2 fair
**Presentation:** 3 good
**Contribution:** 2 fair
**Rating:** 3
**Confidence:** 5

**Summary:**

This paper proposes a sparse training method in federated learning with generative models -- where the goal is to find a binary mask to sparsify the randomly initialized model without training the parameters by borrowing the methodology in [1]. This approach reduces the communication cost, improves storage and inference efficiency, and amplifies privacy when there is a differential privacy mechanism attached to the framework.

[1] Isik, Berivan, et al. "Sparse Random Networks for Communication-Efficient Federated Learning." The Eleventh International Conference on Learning Representations. 2022.

**Strengths:**

The paper successfully adapts the FedPM framework in [1] to generative models and obtains promising empirical results.

The language and the diagrams are clear.

[1] Isik, Berivan, et al. "Sparse Random Networks for Communication-Efficient Federated Learning." The Eleventh International Conference on Learning Representations. 2022.

**Weaknesses:**

- The paper borrows almost all the main ideas from [1]. The only noticeable difference seems to be applying FedPM in [1] to generative models other than classifiers. So, I would expect the authors to highlight this properly and give credit to [1]. Right now, [1] is listed in the related work section very briefly as if there is not much similarity between the two works while the framework is exactly the same.

- Also, I am not sure if there is a need to give the framework in the current paper a new name, PRISM, given that it's actually the same as FedPM but just with a different model/objective?

- I am not sure how the privacy claims follow. The paper cites [1] and [2] for the statement that PRISM (or FedPM) should amplify privacy -- which is correct. But PRISM (or FedPM) alone is not sufficient to have any differential privacy guarantee. They can only amplify privacy when there is an explicit differential privacy mechanism somewhere in the framework. In the experiments section, specifically in Table 1 and Figure 3, both PRISM and PRISM-$\alpha$ are put in the "with privacy" category. Can the authors explain what their privacy mechanism actually is? And how much does the Bernoulli sampling process amplify this privacy?


[1] Isik, Berivan, et al. "Sparse Random Networks for Communication-Efficient Federated Learning." The Eleventh International Conference on Learning Representations. 2022.

[2] Imola, Jacob, and Kamalika Chaudhuri. "Privacy amplification via bernoulli sampling." arXiv preprint arXiv:2105.10594 (2021).

**Questions:**

- Since the proposed method is the same as FedPM [1], it should be mentioned clearly in the revised version.

- What is the reason the authors renamed FedPM as PRISM given that they are the same?

- Where does the differential privacy guarantee come from given that Bernoulli sampling only amplifies privacy but does not introduce any privacy guarantee alone?

---

> ### Author Response · Authors · 2023-11-19
> **[Response 1/1] Thank for your constructive feedbacks!**
>
> **R2-1. Highlight and give credit to FedPM**
>
> As the reviewer suggested, we will more clearly outline the relationship between our research and prior works, including FedPM.
>
> **R2-2. Limited novelty. Why rename FedPM as PRISM?**
>
> Regarding the perceived limited novelty and the renaming of FedPM to PRISM, please see GC2 for a detailed explanation.
>
> As outlined in GC2, our domain faces a significant gap in research that attempts to consider both privacy and communication efficiency simultaneously. This gap, particularly in the context of various losses and frameworks for GANs, results in existing methods underperforming. Our approach addresses this by integrating the MMD loss, which aids in the stable identification of SLT. We also incorporated elements from the decoder structure to improve performance without significantly increasing communication costs. Given these enhancements, we decided to name our method PRISM to reflect its distinctiveness and advancements. Nonetheless, in our revised submission, we will make sure to clarify the evolution from FedPM to PRISM, enabling readers to clearly understand the continuity and progression of ideas.
>
> **R2-3. Differential privacy**
>
> Please see  GC1.

---

> ### Comment · Reviewer_pni9 · 2023-11-23
> **response**
>
> I thank the authors for the rebuttal. The privacy experiments seem rather limited since the privacy cost $\epsilon=50$ is too high. Also, it's not clear how the privacy amplification is used here without knowing the Renyi privacy parameter $\alpha$ since there is no amplification if $d \gamma_{\alpha}(c) > \epsilon$.
>
> I appreciate the authors' efforts in extending FedPM framework to generative models. I believe this could be seen as an important contribution when it's presented in a different way by giving enough credit to the original work and highlighting the necessary modifications.

---

> ### Author Response · Authors · 2023-11-23
> **Response to Review pni9:**
>
> Thank you for your insights and constructive opinions. Your efforts have been a great help in developing our work!
>
> Concerning privacy amplification, we aim to theoretically demonstrate that the privacy bound can be strengthened (or, at the least, remains uncompromised) through the utilization of bernoulli sampling. The degree of amplification may vary depending on different parameters and the model size. However, the crucial aspect is that, given the application of $(\epsilon, \delta)$-dp to $\theta$, there is no scenario where the privacy bound descends below $\epsilon$.

---

> > ### Author Response · Authors · 2023-11-23
> > **Response to using high epsilon**
> >
> > Concerns were raised regarding the experiment results conducted with epsilon=50, as highlighted by Reviewers pni9 and xZ6u. In response, we conducted additional experiments using $\epsilon=9.8$, a value commonly employed in DP-SGD. The outcomes of these experiments can be found in Appendix.A and Tables 6 and 7. Notably, we observed robust and consistent performance from PRISM, while other baselines faced challenges. It's crucial to note that all experiments were conducted with identical $\epsilon$ values for both baselines and PRISM. To eliminate any potential confusion, we have also included information on $\epsilon=9.8$ in GC1.
> >
> > [1] Chen, Dingfan, Tribhuvanesh Orekondy, and Mario Fritz. "Gs-wgan: A gradient-sanitized approach for learning differentially private generators." Advances in Neural Information Processing Systems 33 (2020): 12673-12684.

---

### Official Review · Reviewer_N7hx · 2023-11-01

**Soundness:** 2 fair
**Presentation:** 3 good
**Contribution:** 2 fair
**Rating:** 5
**Confidence:** 3

**Summary:**

Motivated by federated generative models and privacy requirements, this paper proposes a new federated learning scheme. Instead of updating model weights, the authors update a score function for a randomly initialized generator and solely communicate random binary masks. The masks are later used to generate a sub-network for the task. Notably, the network is not trained at all. With the learning scheme, the authors claim it significantly saves communication costs while achieving superior performance in both differentially private (DP) and non-DP settings.

**Strengths:**

1. The problem targeted in this paper is reasonable and practical.

2. The paper is easy to follow.

3. The experimental results are good, though many details are missing.

**Weaknesses:**

1. The novelty is limited. The paper basically applies the prior work [1] in a federated generation setting. However, I did not notice any specialization toward federated learning/generation.

2. Many details and experiments are missing, making it difficult to evaluate the contribution.
    - What kind of architecture is used in the paper?
    - Does the choice of architecture affect the performance?

3. It is unclear how the random mask learning benefits from multiple-round federated learning. It seems to me that it is closer to the one-shot learning. The authors should expand this part and provide more insights.

3. The experiments about differential privacy (DP) are either missing or incomplete. The two most critical parameters in DP, $\varepsilon$ and $\delta$, are missing in the paper. It is also unclear how to apply DP to the proposed method when generating masks. The experiments regarding DP are not meaningful without mentioning privacy budgets. I even doubt the comparison might be unfair.

[1] Sangyeop Yeo, Yoojin Jang, Jy-yong Sohn, Dongyoon Han, and Jaejun Yoo. Can we find strong lottery tickets in generative models? In Proceedings of the AAAI Conference on Artificial Intelligence, volume 37, pp. 3267–3275, 2023.

**Questions:**

1. The performance in some experiments is somehow too surprisingly good to convince me. I am not fully convinced that a randomly initialized model without training can outperform a carefully designed and trained generative model. Let alone the score selection criterion is MMD, which is known to be suboptimal for image generation. Is it because of data partitioning? Could the authors elaborate more on this part?

Overall, though the work is a straightforward application of the strong lottery ticket hypothesis, it still could be valuable to the community. However, the authors have to specify all the details, especially regarding the DP part. I am willing to reconsider my score after the discussion with other reviewers and the authors.

---

> ### Author Response · Authors · 2023-11-19
> **[Response 1/2] Thank you for constructive feedbacks!**
>
> **R1-1. Why can SLT outperform a fully trained generative model? Why is MMD loss effective?**
>
> We interpret your question as probing why the combination of SLT and MMD loss performs so well. We will divide the reviewer’s question into three parts for clarity:
>
> (1) Effectiveness of SLT: Previous research has theoretically shown that in large, randomly initialized models, there often exists a 'strong lottery ticket'--- a subnetwork capable of matching or even surpassing the performance of the full model [1]. To illustrate, consider the analogy of the Library of Babel, where a vast repository of all possible combinations of text exists. In such a library, the probability of finding meaningful books is not negligible. Similarly, in a large enough model, strong and effective subnetworks are likely to exist. This intuitive analogy serves as a foundation for understanding the concept of SLT. Consequently, when such a strong lottery ticket is identified, it has the potential to provide a comparable or sometimes even better performance than fully trained GAN-based models.
>
> Of course, while the theoretical existence of SLTs is significant, the practical challenge lies in the methodology to locate these optimal subnetworks. This requires the careful development of a score function and algorithm tailored to the specific task at hand, which is a highly nontrivial endeavor. (Our contribution lies in this realm, developing a framework that effectively addresses the challenge of creating a federative generative model) Understanding this framework sets the stage to address the reviewer's next question: the effectiveness of MMD loss.
>
> (2) Effectiveness of MMD Loss:
> MMD loss functions by minimizing the mean distance between samples in the Reproducing Kernel Hilbert Space (RKHS). Theoretical evidence suggests that distributions are identical when their statistics in RKHS coincide [2][3]. In practice, this translates to each client computing the L2 distance from the global model to its RKHS statistics during local training, effectively measuring the MMD distance for global statistics. This statistical approach underlies MMD's superiority over traditional GAN-based methods when combined with SLT. This method, as analyzed in the context of WGANs, suggests that MMD, being a distance function, provides more stability in training compared to divergence-based methods like traditional GANs.
>
> This stability is crucial in federated learning environments, where training can be erratic and unstable due to the inherent challenges such as heterogeneous data distributions across clients. The ability of MMD loss to consistently measure distances in RKHS contributes significantly to mitigating these challenges, thereby enhancing model performance and reliability. Thus, our model leverages the combination of SLTs and the stable training dynamics of MMD loss to achieve its notable performance.
>
>
> (3) Comparing GAN and MMD loss: GAN loss is known for its instability and requires extensive regularization and careful training management. To further confirm the advantage of using MMD loss in our setup, we have also explored the possibility of using the GAN loss to find SLT (Figure 10). Due to the inherent training instability of GAN loss, particularly in federated learning settings with data heterogeneity, this approach proved ineffective. Experiments using GAN loss instead of MMD in our PRISM framework resulted in collapsed images, emphasizing GAN loss's unsuitability for SLT in decentralized environments. Nevertheless, as you rightly pointed out, MMD does have limitations in handling large, complex datasets. Exploring alternative frameworks to overcome these challenges is an intriguing direction for future research.
>
> [1] Ramanujan, Vivek, et al. "What's hidden in a randomly weighted neural network?." Proceedings of the IEEE/CVF conference on computer vision and pattern recognition. 2020.
>
> [2] Gretton, Arthur, et al. "A kernel two-sample test." The Journal of Machine Learning Research 13.1 (2012): 723-773.
>
> [3] Gretton, Arthur, et al. "A kernel method for the two-sample-problem." Advances in neural information processing systems 19 (2006).
>
> **R1-2. Limited novelty**
> Please see GC2.
>
> ...
>
> Please refer the remaining responses in the [Responses 2/2].

---

> ### Author Response · Authors · 2023-11-19
> **[Response 2/2] Thank you for constructive feedbacks!**
>
> **R1-3. Which architecture is used? Does the choice matter?**
>
> We have consistently utilized the ResNet-based generator for both our method and all baseline methods to ensure a fair comparison. Our detailed analysis, shown in Figure 4, compares performance while controlling for the number of parameters. The results demonstrate that PRISM outperforms the other baseline methods, even when matched for capacity and network structure. In addition, our method works generally well for different architectures. Basically, SLT just presumes that a dense network is sufficiently overparameterized to find a useful subnetwork. This implies that SLT has robustness with architecture. To validate our argument, we will evaluate architectures frequently used in generative models. When ongoing experiments are completed, we will update you.
>
> **(updated)**
>
> We applied various generator architectures that have been studied in the field of generative models to the architecture of PRISM and compared their performance. Review can find the results in Table.8. We kept all settings, including DP, consistent with our framework. Our default configuration is a ResNet-based generator, and architectures such as DCGAN and SNGAN show robust performance on MNIST but exhibit a slight advantage depending on the architecture in CelebA. PRISM can somewhat follow the performance of the architecture since it extracts a subnetwork from the dense network. Still, it consistently finds a stable SLT for various structures.
>
> | MNIST     |   ResNet   |   DCGAN   |   SNGAN    |
> |:--------------:|:---------------:|:----------------:|:----------------:|
> |  FID         |   34.4595  |    36.9683  |   33.4447   |
> | Precision |   0.3784    |    0.3694    |   0.4652     |
> | Recall      |   0.4213    |    0.4848    |   0.3318     |
> | Density    |   0.1418    |    0.1323    |   0.2047     |
> | Coverage |   0.1588    |    0.1490   |   0.2019     |
>
> |  CelebA   |   ResNet   |   DCGAN   |   SNGAN    |
> |:--------------:|:---------------:|:----------------:|:----------------:|
> |  FID         |   51.0857  |    78.4130  |  38.3009  |
> | Precision |   0.5267    |    0.4540    |   0.6851     |
> | Recall      |   0.0631    |    0.0404    |   0.0925     |
> | Density    |   0.2676    |    0.1916    |   0.5378     |
> | Coverage |   0.2012    |    0.0947    |   0.3155     |
>
>
> **R1-4. How does random mask learning benefit from multiple-round federated learning?**
>
> We interpret your question as probing why we opt for multiple-round learning rather than identifying a Strong Lottery Ticket (SLT) at each client side and then aggregating the final mask in a one-shot manner at the server.
>  To answer this, we draw an analogy between each local dataset in our method and a mini-batch in traditional centralized SGD. In this comparison, one-shot aggregation is akin to optimizing with a very large step size. It is well-known in optimization theory that training with a large step size and few epochs can often lead to local minima. Coming back to the perspective of federated learning, by aggregating the scores (or binary masks) of each client through multiple rounds, our goal is to find the global lottery ticket that is not biased towards specific local dataset. To support our opinion, we conducted an experiment in which we performed single aggregation after overfitting MNIST dataset to 500 local iterations. The results can be found in Appendix.C. We observe that it fails to generate proper images.
>
> **R1-5. Differential privacy**
>
> Please see GC1.

---

> > ### Author Response · Authors · 2023-11-21
> > **updated R1-3**
> >
> > Our response about the choice of architecture is updated in R1-3.

---

> ### Comment · Reviewer_N7hx · 2023-11-23
> **Response to the authors**
>
> I'd like to thank the authors for providing such an informative response. However, as noted by the authors and other reviewers, there are a lot of things to discuss and improve in the paper, such as the difference between prior work and PRISM, privacy analysis, experiments, and some design choices. I am particularly concerned about the privacy part. The authors originally seem to make a wrong statement about the privacy cost of masking ($\varepsilon$ is larger than zero), and the corresponding experiments are also problematic (e.g.,  Table 2 still claims a privacy-preserving property for the proposed method). Even in the GC, the epsilon values seem not convincing. It may take some time to fix thoroughly.
>
> Overall, though I appreciate the interesting application, I'd lean to rejection concerning the current presentation.

---

> > ### Author Response · Authors · 2023-11-23
> > **Response to Review N7hx:**
> >
> > We appreciate that you have provided various perspectives and constructive feedbacks during the revision period!

---

> > > ### Author Response · Authors · 2023-11-23
> > > **Response to using the high epsilon**
> > >
> > > There has been notable concern regarding the experiment results conducted with epsilon=50, as highlighted by Reviewers sDDa and xZ6u. In response to this, we carried out additional experiments using $\epsilon=9.8$, which is frequently used in DP-SGD. The results of these experiments can be found in Appendix.A and Table.6 and 7. Notably, we observed that PRISM robustly performs well, whereas other baselines failed. We would like to notify that all experiments were conducted with identical $\epsilon$ values  for both baselines and PRISM. To avoid any potential confusion, we have also provided information on $\epsilon=9.8$ on GC1. In addition, as can be seen from Table.2 of GS-WGAN[1], they require a large $\epsilon$. This is why we initially picked $\epsilon=50$ in the previous response.
> > >
> > > [1] Chen, Dingfan, Tribhuvanesh Orekondy, and Mario Fritz. "Gs-wgan: A gradient-sanitized approach for learning differentially private generators." Advances in Neural Information Processing Systems 33 (2020): 12673-12684.

---

### Author Response · Authors · 2023-11-13
**Paper revision plan & General Comments**

# Paper revision plan
To ensure transparency and demonstrate our ongoing efforts, we’d like to share the revised paper and our plans for future experiments. As we complete each planned item, we will continuously update this content. Detailed responses to reviewer comments will be posted as soon as our experiments and analysis are completed.

**Our plan:**

* Apply ($\epsilon$, $\delta$) differential privacy before bernoulli sampling (e.g., score values) (All reviewers) : **updated in Appendix.A**
* Test other generator architectures (R1) : **updated (Appendix.C)**
* Apply PRISM with other losses (R4) : **Figure.10 (Appendix.C)**
* Comparison with Multi-FLGAN (R3) : **updated**
* Upload first version of the revised manuscript: **updated**
  * Clearly show our connection with FedPM (R2) :  : **In progress (by Nov.21)**
  * Detailed explanation of experimentals (R2) : **In progress (by Nov.21)**
  * Clarify our contributions and novelty against prior works (R1, R2, R4) : **updated (GC3)**
  * Minor comments : **updated**

# General Comment [1/2]
We thank the reviewers for carefully reviewing our paper and providing constructive feedbacks. Their feedback has greatly improved the clarity and understanding of our work.

**GC1. Differential privacy**

**GC1-1. Explain how PRISM guarantees DP**

The majority of reviewers rightfully pointed out the absence of differential privacy in  bernoulli sampling. Thanks to your feedback, we are pleased to be able to refine our work further. During the revision period, we applied differential privacy before bernoulli sampling (e.g., scores) and updated the results in the Appendix.A (Table.4, 5, Figure.8, 9). For this purpose, we randomized the probability vectors $\theta \in [0,1]^d$ with gaussian differential privacy and clipped them to $\tilde{\theta} \in  [c,1-c]^d$ with small value $0 < c < \frac{1}{2}$. Note that $c$ is 0.1 in our all experiments. We used $\epsilon=50$ and $\delta=0.00001$ as differential privacy parameters, which are consistent with the baselines. The privatized $\tilde{\theta}$ becomes the parameter of the bernoulli distribution and is transformed into a binary mask. Additionally, we regulated the training to ensure that the overall privacy budget does not exceed $\epsilon$, and we applied the same privacy cost regulation to the baselines. Therefore, their performances have been updated, too. All remaining steps are the same as before. From the perspective of PRISM-$\alpha$, which sends both binary mask and non-binary mask scores and binary masks every round, communicating privatized vector $\tilde{\theta} \in [c, 1-c]$ without bernoulli sampling can reduce stochasticity at a small communication cost sacrifice, as mentioned in the paper. PRISM-$\alpha$ also ensures privacy preservation because $\tilde{\theta}$ is basically a dp-guarantee whether passing through bernoulli or not. Therefore, we rearrange PRISM-100 into the privacy category. The overall results still confirm the advantage of our PRISM. We also conducted experiments with frequently used values (epsilon=9.8) such as DP-SGD, and the results can be found in Table.6, 7 of Appendix.A.

**GC1-2. Privacy amplification**

We would discuss about privacy amplification of bernoulli sampling. It is known that bernoulli sampling into privatized information strengthens DP, such as $\epsilon_{amp} \leq \min $ \{$\epsilon, d\gamma_{\alpha}(c)$\}, where $\gamma$ refers binary renyi divergence and $\alpha$ is order of . Note that $d$ limits the privacy amplification when the  model size becomes large [1][2]. Since PRISM assumes that the model size is large enough (due to SLT), we focus on communication efficiency rather than privacy amplification due to bernoulli sampling.

[1] Imola, Jacob, and Kamalika Chaudhuri. "Privacy amplification via bernoulli sampling." arXiv preprint arXiv:2105.10594 (2021).

[2] Isik, Berivan, et al. "Sparse random networks for communication-efficient federated learning." arXiv preprint arXiv:2209.15328 (2022).

...

Please see the rest of our general comment in the following box.

---

> ### Author Response · Authors · 2023-11-19
> **General Comment [2/2]**
>
> **GC2. Lack of Novelty**
> We appreciate the reviewer's observation regarding our method's foundation on the principles outlined in [1] and [2]. It's true that our research can be viewed as an A+B type, where we have combined the concept of Strong Lottery Tickets (SLT) in generative models with the federated learning framework of FedPM. However, the novelty and significance of our work lie in the successful application and adaptation of these ideas to a new and challenging domain – federative generative models.
>
> Federative generative models, particularly in non-IID settings, have been relatively unexplored and pose unique challenges. Traditional federative learning (FL) methods, especially those based on GANs, often suffer from performance issues in these scenarios. Our research addresses this gap by developing a more stable algorithm tailored for federative generative models.
>
> To effectively integrate the concepts from [1] and [2], we encountered and overcame several new challenges. For instance, the mere communication of masks, as suggested in conventional FL, proved insufficient for the complex task of image generation in non-IID FL settings. This led us to devise a novel hybrid score-mask communication strategy, enabling us to finely balance image generation quality with communication load. This strategy, along with other components like weight initialization and ternary quantization, was pivotal in adapting the SLT concept to the unique demands of federative generative models.
>
> Furthermore, our work pioneers the application of these ideas in the context of challenging non-IID federative generative model settings. We are the first to demonstrate successful image generation on datasets like CelebA under distributed and privacy-conscious settings, which marks a significant advancement in this field.
>
> In summary, while our research builds upon existing work, the value of our contribution lies in the unique application, adaptation, and overcoming of specific challenges in a relatively unexplored domain. This not only validates the A+B approach but also extends it by adding new dimensions and insights, thereby advancing the field of federative generative models.
>
> We will clarify these points in our manuscript and state the relationship to the prior works more clearly as the reviewers suggested. We hope that these new findings as well as new technical contributions could be acknowledged by the reviewers.
>
> [1] Isik, Berivan, et al. "Sparse random networks for communication-efficient federated learning." arXiv preprint arXiv:2209.15328 (2022).
> [2] Yeo, Sangyeop, et al. "Can we find strong lottery tickets in generative models?." Proceedings of the AAAI Conference on Artificial Intelligence. Vol. 37. No. 3. 2023.

---

### Meta-Review · Area_Chair_j6Q2 · 2023-12-06

**Metareview:**

This paper proposes a new federated learning (FL) algorithm for generative models. The algorithm is based on the so-called Strong Lottery Ticket (SLT) hypothesis: a random large NN contains a small subnetwork that already solves the desired task (Frankle & Carbin 2018). In the non-private setting, efficient algorithms for finding such a subnetwork is also known, including the edge-popup (EP) algorithm of (Ramanujan et al., 2020). The main contribution of this paper is to adapt this to the FL setting. Roughly speaking, the EP algorithm computes the score of each edge from each sample / client, aggregate the scores and then keep only certain top-score edges in the network. This can be naturally adapted to the FL setting (where we can just FL aggregation). The authors also propose an improvement here where each client sends a binary mask (created randomly from the scores) instead of the actual scores. This helps reduce the communication further. The protocol also uses maximum mean discrepancy (MMD) loss to help stabilize the training.

# Strengths

- The problem is of practical interest and empirical results suggest that the protocols are practical.

# Weaknesses

- Limited novelty: It turns out that the entire main algorithm I described above is more or less presented already in a work by Isik et al. (ICLR'23). (The original submission did not discuss this at all until one of the reviewer pointed it out.) Given this, the main contribution seems to be just to use MMD loss instead of GAN loss.

- Differential privacy analysis: The privacy claims of the paper are confusing. For example, they say that they get privacy from random mask sampling, but this is not strictly true for differential privacy (which the paper clearly alludes to). The authors tried to mitigate this by adding a brief section during the rebuttal period, but this section is quite haphazard and the differential privacy claims remain unclear.

**Justification For Why Not Higher Score:**

The novelty is clearly too little a contribution for ICLR. Confusing privacy claims also do not help.

**Justification For Why Not Lower Score:**

N/A

---

### Decision · Program_Chairs · 2024-01-16

Reject